# Improving Differentiable Neural Architecture Search by Encouraging Transferability

**Parth Sheth**[*]
University of Pennsylvania
parthfour@gmail.com

**Pengtao Xie**[†]
University of California San Diego
p1xie@eng.ucsd.edu

## Abstract

Differentiable neural architecture search methods are increasingly popular due to their computational efficiency. However, these methods have unsatisfactory generalizability and stability. Their searched architectures are often degenerate with a dominant number of skip connections and perform unsatisfactorily on test data. Existing methods for solving this problem have a variety of limitations, such as cannot prevent the happening of architecture degeneration, being excessively restrictive in setting the number of skip connections, etc. To address these limitations, we propose a new approach for improving the generalizability and stability of differentiable NAS, by developing a transferability-encouraging tri-level optimization framework which improves the architecture of a main model by encouraging good transferability to an auxiliary model. Our framework involves three stages performed end-to-end: 1) train network weights of a main model; 2) transfer knowledge from the main model to an auxiliary model; 3) optimize the architecture of the main model by maximizing its transferability to the auxiliary model. We propose a new knowledge transfer approach based on matching quadruple relative similarities. Experiments on several datasets demonstrate the effectiveness of our method.

## 1 Introduction

Neural architecture search (NAS) (Zoph & Le, 2017; Liu et al., 2018b; Cai et al., 2019; Liu et al., 2019a; Pham et al., 2018; Real et al., 2019), which aims to search for highly-performant neural architectures automatically, finds broad applications. Among various NAS methods, differentiable search methods (Liu et al., 2018b; Cai et al., 2019; Chen et al., 2019; Xu et al., 2020) gain increasing popularity due to their computational efficiency. In differentiable NAS, architectures are represented as differentiable variables and are learned using gradient descent.

While differentiable NAS is computationally efficient, its generalizability and stability has been challenged in many works (Zela et al., 2019; Chu et al., 2020a; 2019; Zhou et al., 2020a; Chen & Hsieh, 2020): the searched architecture is degenerate with a dominant number of skip connections; while having good performance on validation data, it performs unsatisfactorily on test data. For example, Zela et al. (2019) identified 12 NAS benchmarks based on four search spaces where architectures searched by standard DARTS (Liu et al., 2019a) (a differentiable NAS method) have poor performance on test data of CIFAR-10, CIFAR-100, and SVH. A variety of approaches (Zela et al., 2019; Chu et al., 2020a; 2019; Zhou et al., 2020a; Chen & Hsieh, 2020; Chen et al., 2019; Liang et al., 2020; Wang et al., 2021) have been proposed to improve the generalizability and stability of differentiable NAS methods. These methods have various limitations, such as cannot improve search algorithms to prevent degenerate architectures from occurring (Zela et al., 2019), cannot explicitly maximize the generalization performance of architectures (Chu et al., 2020a; 2019), cannot broadly explore search spaces (Zhou et al., 2020a; Chen & Hsieh, 2020), requiring extensive tuning of the number of skip connections (Chen et al., 2019; Liang et al., 2020), etc. As a result, their effectiveness in improving differentiable NAS is less satisfactory.

To address these limitations, we propose a new approach for improving the generalizability and stability of differentiable NAS methods. Specifically, we develop a transferability-encouraging tri-level

---

[*]The work was done while visiting UCSD.
[†]Corresponding author

optimization (TETLO) framework, which improves the architecture of a main model by encouraging effective knowledge transfer to an auxiliary model. Intuitively, to train a better auxiliary model, the main model needs to generate accurate knowledge (which is used to train the auxiliary); to generate accurate knowledge, the main model needs to improve its architecture (which is used to generate knowledge). Combining these two steps together, we conjecture that improving the auxiliary drives the main model to improve its architecture (empirical evidence is provided in Figure 2). Our method is also motivated by the theoretical analysis in (Liu et al., 2019b) that good transferability improves generalization performance.

In our framework, a main model and an auxiliary model have architectures and network weights to learn. Learning consists of three stages. In the first stage, we train the network weights of a main model while temporarily fixing its architecture. In the second stage, we leverage the main model trained in the first stage to help train an auxiliary model via transfer learning. To capture high-order data relationships, we propose a new knowledge transfer approach based on matching quadruple relative similarities (QRS), where the main model generates QRS relationships, e.g., the similarity between data example $x$ and $y$ is larger than that between $w$ and $z$. Then the auxiliary model is trained by fitting these QRS relationships. In the third stage, we use validation performance of the auxiliary model as a measure of transferability from the main model to the auxiliary model and update the architecture of the main model by maximizing the transferability. The three stages are performed end-to-end in a three-level optimization framework.

The major contributions of this work include:

- We propose a transferability-encouraging tri-level optimization (TETLO) framework to improve the generalizability and stability of differentiable NAS methods.
- We propose a new knowledge transfer approach based on matching quadruple relative similarities.
- We perform various experiments which demonstrate the effectiveness of our method.

## 2 RELATED WORKS

### 2.1 NEURAL ARCHITECTURE SEARCH

The goal of neural architecture search (NAS) is to automatically identify highly-performing neural architectures that can potentially surpass human-designed ones. The research of NAS has made considerable progress in the past few years. Early NAS approaches (Zoph & Le, 2017; Pham et al., 2018; Zoph et al., 2018) are based on reinforcement learning (RL), where a policy network learns to generate high-quality architectures by maximizing validation accuracy (used as reward). These approaches are conceptually simple and can flexibly perform search in any search space. In differentiable search methods (Cai et al., 2019; Liu et al., 2019a; Xie et al., 2019), each candidate architecture is a combination of many building blocks. Combination coefficients represent the importance of building blocks. Architecture search amounts to learning these differentiable coefficients, which can be conducted using differentiable optimization algorithms such as gradient descent. Another paradigm of NAS methods (Liu et al., 2018b; Real et al., 2019) are based on evolutionary algorithms (EA). In these approaches, architectures are considered as individuals in a population. Each architecture is associated with a fitness score representing how good this architecture is. Architectures with higher fitness scores have higher odds of generating offspring (new architectures), which replace old architectures that have low-fitness scores. Our proposed method can be applied to any differentiable NAS methods. Recently, meta-NAS methods (Elsken et al., 2020; Lian et al., 2019) have been proposed for fast adaptation of neural architectures. Our method differs from meta-NAS approaches in the following aspects. First, our method aims at improving an architecture by encouraging it to have good transferability to an auxiliary model, while meta-NAS focuses on adapting a meta architecture to different task-specific architectures. Second, our method transfers knowledge from a main architecture to an auxiliary model using our newly proposed quadruple relative similarity matching mechanism while meta-NAS adapts a meta architecture to task-specific architectures via gradient descent update. Third, our method searches for architectures on a single dataset while meta-NAS operates on a collection of meta-training tasks (each with a training and validation set).

### 2.2 IMPROVE THE GENERALIZABILITY AND STABILITY OF DIFFERENTIABLE NAS

Various methods have been proposed for improving the generalizability and stability of differentiable NAS methods. Zela et al. (2019) proposed an early stopping approach based on eigenvalues of validation loss' Hessian matrix. This method early stops search algorithms when degenerate architectures occur, but can not improve search algorithms to prevent degenerate architectures from oc-

curring. Chu et al. (2020a) leverage auxiliary skip connections to encourage a fairer competition for all operations. Chu et al. (2019) encourage each operation's architectural weight to be independent of others. These methods focus on reducing the number of skip connections, but lack a mechanism of encouraging searched architectures to learn representations that can be well transferred to other models. In contrast, our method explicitly improves the transferability of representations learned by searched architectures via minimizing the validation loss of an auxiliary model trained on transferred representations. Zhou et al. (2020a) proposed a path-depth-wise regularization approach. Chen & Hsieh (2020) developed a perturbation-based regularization method. In these regularization-based methods, it is difficult to choose a proper level of regularization strength. As theoretically analyzed in (Xie et al., 2017), an excessively strong regularization will lead to underfitting while an insufficient regularization will lead to overfitting. Finding the sweeping point between underfitting and overfitting needs extensive tuning, which is time-consuming and computationally expensive. Our method does not have this limitation since it is not regularization-based and therefore does not need to find the optimal level of regularization strength. Some methods in (Chen et al., 2019; Liang et al., 2020) constrain the number of skip connections to a fixed number (typically 2), which might be too restrictive and consequently hurts performance. In contrast, our method does not have hard constraints on the number of skip connections. Wang et al. (2021) identified that magnitude-based architecture selection approaches hurt generalization and proposed a perturbation-based architecture selection method to address this problem. Our proposed method, which focuses on training better weights of the supernet, is orthogonal to (Wang et al., 2021) and can be applied together with (Wang et al., 2021) to improve the generalizability and stability of differentiable NAS.

## 2.3 TRANSFER LEARNING

Existing transfer learning (TL) methods (Pratt, 1993; Mihalkova et al., 2007; Niculescu-Mizil & Caruana, 2007; Pan & Yang, 2009; Luo et al., 2017; Zhuang et al., 2020) aim to improve performance of a target task by transferring knowledge from a source task. Various transfer strategies have been proposed, based on reweighting source data to better match target distribution (Huang et al., 2006; Jiang & Zhai, 2007; Sugiyama et al., 2007; Foster et al., 2010; Moore & Lewis, 2010; Axelrod et al., 2011; Wang et al., 2017b; Ngiam et al., 2018), transforming source and target data into a common feature space (Borgwardt et al., 2006; Pan et al., 2008; 2010; Duan et al., 2012; Long et al., 2013; Wang et al., 2017a), extracting domain-invariant representations based on adversarial learning (Ganin et al., 2016; Long et al., 2017; Tzeng et al., 2017; Hoffman et al., 2018; Shen et al., 2018; Zhang et al., 2019), regularizing target models using source models (Luo et al., 2008; Duan et al., 2009; Zhuang et al., 2009; Tommasi et al., 2010; Duan et al., 2012), etc. Existing methods improve a target model by absorbing knowledge from a source model. In our work, we take an opposite direction, which improves a main model by encouraging good transferability to an auxiliary model.

## 3 METHODS

In this section, we present the transferability-encouraging tri-level optimization framework (Figure). Given a main task $m$ (e.g., image classification (He et al., 2016a)) with a main dataset $D_m$, we aim to learn a

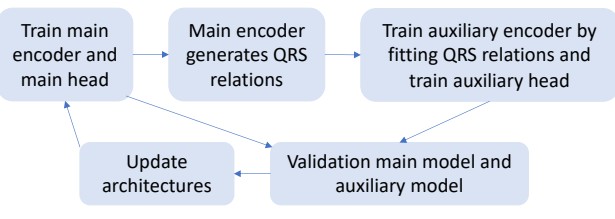

Figure 1: Overview of the TETLO framework.

high-performance main model to conduct this task. The main model has a learnable architecture $M$, a data encoder $E_m$, and a task-specific head $H_m$. To prevent $M$ from being degenerate, we encourage it to have good transferability to an auxiliary model. Via transfer learning, we leverage the main model to train an auxiliary model in an auxiliary task with an auxiliary dataset $D_a$. The auxiliary model has a learnable architecture $A$, an encoder $E_a$, and a task-specific head $H_a$. We use the validation performance of the auxiliary model to measure main-to-auxiliary transferability. Since the auxiliary model is trained with the help of the main model, good validation performance of the auxiliary model indicates strong generalization power of the main model. We optimize the architecture of the main model by maximizing the auxiliary's validation performance.

### 3.1 A THREE-LEVEL OPTIMIZATION BASED FRAMEWORK

Learning consists of three stages which are performed end-to-end. In the first stage, the main model trains its network weights with its architecture temporarily fixed. In the second stage, the auxiliary model is trained with the help of the main model via transfer learning. In the third stage, we measure

main-to-auxiliary transferability, which is the validation performance of the auxiliary model, and update the architecture of the main model by maximizing transferability.

**Stage I.** In this stage, we perform train the main encoder $E_m$ and main head $H_m$ on $D_m^{(tr)}$ with the architecture $M$ tentatively fixed, where $D_m^{(tr)}$ is the training set of $D_m$. Let $L_m$ denote a training loss (e.g., cross entropy for classification). This stage amounts to solving the following problem:

$$E_m^*(M), H_m^*(M) = \text{argmin}_{E_m,H_m} L_m(M, E_m, H_m, D_m^{(tr)}). \tag{1}$$

$M$ is needed to calculate $L_m$. However, it should not be updated via minimizing $L_m$. Otherwise, a trivial solution will be produced where $M$ can perfectly overfit $D_m^{(tr)}$ but will yield poor performance on test examples. $E_m^*(M)$ and $H_m^*(M)$ denote that optimal solutions $E^*$ and $H^*$ are functions of $M$: a different $M$ will result in a different training loss, which renders $E_m^*$ and $H_m^*$ to change accordingly.

**Stage II.** In the second stage, we use transfer learning to train the auxiliary model with the help of the main model. Previous knowledge transfer approaches (Romero et al., 2014; You et al., 2017; Pham et al., 2020; Chen et al., 2020a) are limited to capturing low-order ($<= 3$) relationships among data examples during knowledge transfer and fail to explore high-order relationships. To address this problem, we propose a new transfer learning approach, which can explore fourth-order data relationships, based on matching quadruple relative similarity. Given two randomly sampled data pairs $(x,y)$ and $(z,w)$ from $D_a^{(tr)}$, we use the main model to determine which pair has more similar data examples. For each pair $(x,y)$, the main encoder (including $E_m^*(M)$ and $M$) trained in the first stage is used to encode the two examples in this pair; then cosine similarity $c(x,y;E_m^*(M),M)$ is calculated between the two encodings. Given two pairs $(x,y)$ and $(z,w)$, a quadruple relative similarity relation is about whether $c(x,y;E_m^*(M),M) > c(z,w;E_m^*(M),M)$. Given quadruple relative similarity relations generated by the main encoder, we train the auxiliary encoder (including $E_a$ and $A$) to match these relations. If $c(x,y;E_m^*(M),M) > c(z,w;E_m^*(M),M)$, then it is required that $c(x,y;E_a,A) > c(z,w;E_a,A)$, where $c(x,y;E_a,A)$ is the cosine similarity between $x$ and $y$ which are encoded using the auxiliary encoder. Since each quadruple involves four examples, our method is able to capture data relationships with an order of four. In this stage, we solve the following problem:

$$E_a^*(M,A), H_a^*(A) = \text{argmin}_{E_a,H_a} L_a(A, E_a, H_a, D_a^{(tr)})$$
$$s.t. \ \forall(x,y,z,w), [c(x,y;E_m^*(M),M) > c(z,w;E_m^*(M),M)] \rightarrow [c(x,y;E_a,A) > c(z,w;E_a,A)]$$

**Stage III.** In the third stage, we measure "transferability" from the main model to the auxiliary model using the validation performance of the auxiliary model. Since the auxiliary model is trained using knowledge generated by the main model, the auxiliary's validation performance reflects the quality of main-to-auxiliary "transferability". A smaller validation loss indicates the knowledge generated by the main model has better quality, which implies that the main model has better generalization ability. We update the architecture of the main model by minimizing the auxiliary's validation loss. In addition, we can also measure the validation loss of the main model and update its architecture by minimizing the main model's validation loss as well. This amounts to solving the following problem:

$$\min_{M,A} L_m(M, E_m^*(M), H_m^*(M), D_m^{(val)}) + \gamma L_a(A, E_a^*(M,A), H_a^*(A), D_a^{(val)}), \tag{2}$$

where $D_m^{(val)}$ and $D_a^{(val)}$ is the validation set of $D_m$ and $D_a$ respectively. $\gamma$ is a tradeoff parameter.

**Three-level optimization framework.** Putting these pieces together, we have the following transferability-encouraging tri-level optimization (TETLO) problem:

$$\min_{M,A} L_m(M, E_m^*(M), H_m^*(M), D_m^{(val)}) + \gamma L_a(A, E_a^*(M,A), H_a^*(A), D_a^{(val)})$$
$$s.t.$$
$$E_a^*(M,A), H_a^*(A) = \text{argmin}_{E_a,H_a} L_a(A, E_a, H_a, D_a^{(tr)})$$
$$s.t. \ \forall(x,y,z,w), [c(x,y;E_m^*(M),M) > c(z,w;E_m^*(M),M)] \rightarrow [c(x,y;E_a,A) > c(z,w;E_a,A)] \tag{3}$$

$$E_m^*(M), H_m^*(M) = \text{argmin}_{E_m,H_m} L_m(M, E_m, H_m, D_m^{(tr)})$$

**Continuous relaxation.** The constraints in Eq.(3) are not amenable for optimization. We perform a relaxation of them to make optimization easier. To satisfy the constraint $[c(x,y;E_m^*(M),M) > c(z,w;E_m^*(M),M)] \rightarrow [c(x,y;E_a,A) > c(z,w;E_a,A)]$ in Eq.(3), we can encourage $(c(x,y;E_m^*(M),M) - c(z,w;E_m^*(M),M))(c(x,y;E_a,A) - c(z,w;E_a,A))$ to be positive. Under this relaxation, the optimization problem at stage II becomes as follows, where $\lambda$ is a tradeoff parameter:

$$E_a^*(M,A), H_a^*(A) = \text{argmin}_{E_a,H_a} L_a(A, E_a, H_a, D_a^{(tr)}) +$$
$$\lambda \sum_{x,y\in D_a^{(tr)}} \max(0, -(c(x,y;E_m^*(M),M) - c(z,w;E_m^*(M),M))(c(x,y;E_a,A) - c(z,w;E_a,A))) \tag{4}$$

Table 1: Test errors of CIFAR-10 (C10) and CIFAR-100 (C100) under four search spaces. †: the number of initial channels is 16 and the number of cells is 8. ‡: the number of initial channels is 36 and the number of cells is 20. [1] (Liu et al., 2019a), [2] (Zela et al., 2019), [3] (Zela et al., 2019), [4] (Chu et al., 2020a), [5] (Chen & Hsieh, 2020), [6] (Maninis et al., 2019).

| Data | Space | DARTS [1] | RDARTS-L2 [2] | DARTS-ES [3] | DARTS- [4] | SDARTS-ADV [5] | MTL [6] | Ours |
|------|-------|-----------|---------------|--------------|------------|----------------|---------|------|
| C10 | S1† | 4.69 | 3.46 | 3.93 | 3.34 | 3.26 | 4.39 | **3.12** |
| | S1‡ | 3.84 | 2.78 | 3.01 | 2.68 | 2.73 | 3.07 | **2.51** |
| | S2 | 5.54 | 3.31 | 4.07 | 4.03 | 3.11 | 4.11 | **2.94** |
| | S3 | 3.92 | 2.51 | 3.55 | 2.95 | 3.07 | 4.61 | **2.44** |
| | S4 | 8.33 | 3.56 | 4.69 | 4.14 | 3.49 | 6.05 | **3.32** |
| C100 | S1 | 29.46 | 24.25 | 28.37 | 22.41 | 22.33 | 27.85 | **21.63** |
| | S2 | 26.05 | 22.24 | 23.25 | 21.61 | 20.56 | 24.25 | **19.72** |
| | S3 | 28.90 | 23.99 | 23.73 | 21.13 | 21.08 | 21.40 | **20.35** |
| | S4 | 22.85 | 21.94 | 21.26 | 21.55 | 21.25 | 22.97 | **20.51** |

**Differentiable parameterization of architecture variables.** Following (Liu et al., 2019a), we parameterize architectures in a differentiable way. Each candidate block in an architecture is associated with an importance weight. Architecture search amounts to learning these weights using gradient methods. After learning, blocks with top weights compose an architecture.

**Optimization algorithm.** To solve the problem in Eq.(3) (with relaxation), we used a standard algorithm developed in (Liu et al., 2019a), which is broadly used in many previous works and demonstrated to be effective in the literature. The convergence analysis of this algorithm has been given in many works (Ghadimi & Wang, 2018; Grazzi et al., 2020; Ji et al., 2021; Liu et al., 2021; Yang et al., 2021). The optimization algorithm is not the focus or contribution of our work. Please see Section H for the algorithm.

**Reduce computational and memory costs.** To reduce the computational and memory costs, we leverage the following methods. First, we perform architecture tying and weight parameter tying between the main encoder and auxiliary encoder. For both the main and auxiliary model, we set them to be a stack of 7 cells, sharing the same architecture as the first 7 cells in DARTS. Parameter tying is performed by making the weight parameters of the first six cells in the two models be the same. For the seventh cell, the main and auxiliary models have different weight parameters. Second, following (Liu et al., 2019a), we use one-step gradient descent update to approximate optimal solutions and use finite difference approximation to approximate matrix-vector multiplications. Third, we reduce the frequencies of updating (including calculating hypergradients) the architecture $M$. It is updated every 5 mini-batches (i.e., iterations) instead of on every mini-batch. We empirically found doing this greatly reduced computational costs without significantly sacrificing accuracy. The rest of parameters were updated on every mini-batch.

## 4 EXPERIMENTS

In this section, we present experimental results. We compare our method with state-of-the-art methods 1) on search spaces designed by (Zela et al., 2019) for measuring generalizability and stability of differentiable NAS methods; 2) on CIFAR-100, CIFAR-10, and ImageNet under search spaces of DARTS, P-DARTS, PC-DARTS, and PR-DARTS. Please refer to the appendix for detailed hyperparameter settings and additional results.

The main and auxiliary tasks are both image classification, but are conducted using different models. Following (Liu et al., 2019a), each experiment consists of 1) an architecture search phrase which searches for an optimal cell by minimizing the loss on a validation set, and 2) an architecture evaluation phrase which composes multiple copies of the searched cell into a larger network and trains the network on the combination of training and validation data, and evaluates it on a test set.

### 4.1 DATASETS

Three datasets are used in the experiments, including ImageNet (Deng et al., 2009), CIFAR-100 (Krizhevsky et al., 2009), and CIFAR-10 (Krizhevsky & Hinton, 2010). ImageNet is split into a training set and a test set with 1.2M and 50K images respectively. CIFAR-100 is split into training, validation, and test sets, with 25K, 25K, and 10K images respectively. So is CIFAR-10. These three datasets have 1000, 100, and 10 classes respectively.

### 4.2 EXPERIMENTAL SETTINGS

The head of the main model for CIFAR-10, CIFAR-100, and ImageNet is set to a 10-way, 100-way, and 1000-way linear classifier respectively. For each dataset, its training set is used as $D_m^{(\text{tr})}$ and

$D_a^{(tr)}$; its validation set is used as $D_m^{(val)}$ and $D_a^{(val)}$. Tradeoff parameters $\gamma$ and $\lambda$ are set to 1 and 0.1 respectively. We compare with the following baselines: 1) multi-task learning (MTL) (Maninis et al., 2019): main and auxiliary models are trained by minimizing the weighted sum of their losses; 2) encoder sharing (ES) (Kokkinos, 2017): main and auxiliary models share the same encoder; 3) auxiliary-to-main transfer (ATT) (He et al., 2016b): first train an auxiliary model on $D_a^{(tr)}$ and $D_a^{(val)}$, then train a main model by transferring knowledge from auxiliary. We report mean and standard deviation of 10 random runs following (Liu et al., 2019a).

## 4.3 Evaluation of generalizability and stability on four search spaces

We evaluate our method on four architecture search spaces designed by (Zela et al., 2019) for measuring the generalizability and stability of differentiable NAS methods. Following RobustDARTS-L2 (Zela et al., 2019), we set the number of cells to 8 and the number of initial channels to 16 for CIFAR-10, and set cell number to 8 and initial channel number to 16 for CIFAR-100. Following DARTS (Liu et al., 2019a) and RobustDARTS (Zela et al., 2019), the search in each method was run 4 times with different random seeds independently. Each searched architecture is retrained from scratch for a few epochs. The one (after retraining) yielding the best validation accuracy is selected. Architecture evaluation is performed on the selected architecture to generate the final result. Table 1 shows the results. Our method achieves lower test errors on CIFAR100/10 under these four spaces than baselines, which demonstrates our method has better generalizability and stability than baselines. The reason is: our method explicitly encourages the architecture $M$ of a main model to have good transferability to an auxiliary model; in this way, $M$ is learned to be effective not only for the main task, but also for the auxiliary task, which results in better generalizability.

Table 2: Classification error (%) on CIFAR-100 test set, number of model weights (millions), and search cost (GPU days). TETLO-darts1st: TETLO is applied to DARTS-1st. Similar meanings hold for other such notations. Results marked with ∗ are obtained from DARTS⁻ (Chu et al., 2020a) and $\beta$-DARTS (Ye et al., 2022). Methods marked with † were rerun 10 times. Search cost is measured by GPU days on a Nvidia 1080Ti.

| Method | Error(%) | Param | Cost |
|---|---|---|---|
| *ResNet (He et al., 2016a) | 22.10 | 1.7 | - |
| *DenseNet (Huang et al., 2017) | 17.18 | 25.6 | - |
| *PNAS (Liu et al., 2018a) | 19.53 | 3.2 | 150 |
| *ENAS (Pham et al., 2018) | 19.43 | 4.6 | 0.5 |
| *AmoebaNet (Real et al., 2019) | 18.93 | 3.1 | 3150 |
| *GDAS (Dong & Yang, 2019) | 18.38 | 3.4 | 0.2 |
| *R-DARTS (Zela et al., 2020) | 18.01±0.26 | - | 1.6 |
| *DARTS⁻ (Chu et al., 2020a) | 17.51±0.25 | 3.3 | 0.4 |
| *DropNAS (Hong et al., 2020) | 16.95±0.41 | 4.4 | 0.7 |
| *DOTS (Gu et al., 2021) | 16.48±0.13 | 4.1 | 0.3 |
| *$\beta$-DARTS (Ye et al., 2022) | 16.24 ±0.22 | 3.8 | 0.4 |
| *Darts2nd (Liu et al., 2019a) | 20.58±0.44 | 3.1 | 4.0 |
| ATT-darts2nd (He et al., 2016b) | 19.62±0.17 | 3.2 | 4.4 |
| ES-darts2nd (Kokkinos, 2017) | 19.34±0.21 | 3.1 | 4.0 |
| MTL-darts2nd (Maninis et al., 2019) | 18.71±0.26 | 3.1 | 4.1 |
| TETLO-darts2nd (ours) | **17.21**±0.10 | 3.0 | 4.0 |
| *Pdarts (Chen et al., 2019) | 17.42±0.14 | 3.6 | 0.3 |
| ATT-pdarts (He et al., 2016b) | 17.37±0.16 | 3.6 | 0.5 |
| ES-pdarts (Kokkinos, 2017) | 17.35±0.21 | 3.7 | 0.3 |
| MTL-pdarts (Maninis et al., 2019) | 17.22±0.24 | 3.6 | 0.4 |
| TETLO-pdarts (ours) | **16.05**±0.08 | 3.6 | 0.3 |
| †Pcdarts (Xu et al., 2020) | 17.96±0.15 | 3.9 | 0.1 |
| ATT-pcdarts (He et al., 2016b) | 17.71±0.19 | 4.2 | 0.2 |
| ES-pcdarts (Kokkinos, 2017) | 17.82±0.16 | 3.9 | 0.1 |
| MTL-pcdarts (Maninis et al., 2019) | 17.85±0.23 | 4.0 | 0.2 |
| TETLO-pcdarts (ours) | **16.42**±0.07 | 3.8 | 0.1 |
| †Prdarts (Zhou et al., 2020b) | 16.48±0.06 | 3.4 | 0.2 |
| ATT-prdarts (He et al., 2016b) | 17.35±0.14 | 3.4 | 0.4 |
| ES-prdarts (Kokkinos, 2017) | 16.73±0.27 | 3.5 | 0.2 |
| MTL-prdarts (Maninis et al., 2019) | 17.46±0.10 | 3.3 | 0.3 |
| TETLO-prdarts (ours) | **15.81**±0.06 | 3.4 | 0.2 |

## 4.4 CIFAR-100 and CIFAR-10 results under search spaces of DARTS, P-DARTS, PC-DARTS, and PR-DARTS

For the main encoder architecture, we experimented with the search spaces in DARTS (Liu et al., 2019a), P-DARTS (Chen et al., 2019), PC-DARTS (Xu et al., 2020), and PR-DARTS (Zhou et al., 2020b). During architecture search on CIFAR-100 and CIFAR-10, the architecture is a stack of 8 cells, each consisting of 7 nodes. Initial channel number was set to 16. The rest of hyperparameters follow those in DARTS, P-DARTS, PC-DARTS, and PR-DARTS.

In Table 2, we compare different NAS methods on CIFAR-100. From these results, we observe the following. First of all, when our method TETLO is applied to DARTS, P-DARTS, PC-DARTS,

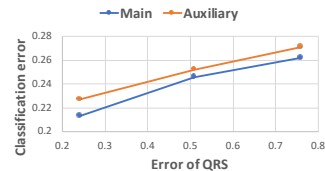

Figure 2: How validation errors of the auxiliary and main models change with the error of QRS.

and PR-DARTS, classification errors on test data are significantly reduced (significance test results are in Appendix R), which shows that encouraging main-to-auxiliary transferability can improve the generalizability of the main model. Table 18 shows that architectures searched by our method have

fewer skip connections than baselines, which demonstrates that our method has better stability. By minimizing the validation loss of the auxiliary trained on knowledge (quadruple relative similarities) generated by the main model, more accurate knowledge can be generated. By promoting the fidelity of generated knowledge, the generalizability and stability of the main model is improved.

For the above analysis, we provide empirical evidence. To show that minimizing auxiliary's validation loss can yield more accurate quadruple relative similarity (QRS) relationships, we create QRS relationships with different levels of accuracy. Given $(x, y)$ and $(z, w)$, if $x$ and $y$ have the same class label and $z$ and $w$ have different class labels, then the similarity $c(x, y)$ between $x$ and $y$ is considered to be larger than the similarity $c(z, w)$ between $z$ and $w$. First, we create a set of fully accurate QRS relationships. Then we randomly sample 25%, 50%, and 75% quadruples and make their relative similarity relationships incorrect, i.e., changing $c(x, y) > c(z, w)$ to $c(x, y) < c(z, w)$. By doing this, we generate three sets of QRS with decreasing accuracy. Using the three sets as constraints, we train three auxiliary models. Figure 2 shows that classification errors on a validation set increase as the error of QRS relationships increases. These results show that the fidelity of knowledge is positively correlated to auxiliary's classification accuracy. Therefore, minimizing auxiliary's validation loss can yield more reasonable knowledge.

To show that promoting knowledge' fidelity can improve the generalizability of the main model, we train main models by fitting QRS relationships that have different accuracy. Then we measure classification errors of these models. Figure 2 shows that a main model trained using more accurate QRS relationships achieves lower classification error. This indicates the fidelity of knowledge is positively correlated with the classification accuracy of the main model. Therefore, promoting knowledge' fidelity can improve the generalizability of the main model.

Table 3: Results on CIFAR-10, including classification error (%) on the test set, number of model weights (millions), and search cost (GPU days). * denotes that the results are taken from DARTS$^-$ (Chu et al., 2020a), NoisyDARTS (Chu et al., 2020b), DrNAS (Chen et al., 2020b), $\beta$-DARTS (Ye et al., 2022), and AGNAS (Sun et al., 2022). The rest notations are the same as those in Table 2 in the main paper.

| Method | Error(%) | Param(M) | Cost |
|---|---|---|---|
| *DenseNet (Huang et al., 2017) | 3.46 | 25.6 | - |
| *ENAS (Pham et al., 2018) | 2.89 | 4.6 | 0.5 |
| *NASNet-A (Zoph et al., 2018) | 2.65 | 3.3 | 1800 |
| *AmoebaNet-B (Real et al., 2019) | 2.55±0.05 | 2.8 | 3150 |
| *R-DARTS (Zela et al., 2020) | 2.95±0.21 | - | 1.6 |
| *ASAP (Noy et al., 2020) | 2.68±0.11 | 2.5 | 0.2 |
| *SDARTS (Chen & Hsieh, 2020) | 2.61±0.02 | 3.3 | 1.3 |
| *DARTS+PT (Wang et al., 2021) | 2.61±0.08 | 3.0 | 0.8 |
| *DropNAS (Hong et al., 2020) | 2.58±0.14 | 4.1 | 0.6 |
| *FairDARTS (Chu et al., 2019) | 2.54 | 3.3 | 0.4 |
| *DrNAS (Chen et al., 2020b) | 2.54±0.03 | 4.0 | 0.4 |
| *ISTA-NAS (Yang et al., 2020) | 2.54±0.05 | 3.3 | 0.1 |
| *MiLeNAS (He et al., 2020) | 2.51±0.11 | 3.9 | 0.3 |
| *GAEA (Li et al., 2021) | 2.50±0.06 | - | 0.1 |
| *PDARTS-ADV (Chen & Hsieh, 2020) | 2.48±0.02 | 3.4 | 1.1 |
| *GTN(F=128) (Such et al., 2019) | 2.42±0.03 | 97.9 | 0.67 |
| *DOTS (Gu et al., 2021) | 2.49±0.06 | 3.5 | 0.3 |
| *$\beta$-DARTS (Ye et al., 2022) | 2.53±0.08 | 3.8 | 0.4 |
| *AGNAS (Sun et al., 2022) | 2.53±.003 | 3.6 | 0.4 |
| *Darts2nd (Liu et al., 2019a) | 2.76±0.09 | 3.3 | 4.0 |
| ATT-darts2nd (He et al., 2016b) | 2.78±0.05 | 3.4 | 4.4 |
| ES-darts2nd (Kokkinos, 2017) | 2.76±0.14 | 3.4 | 4.0 |
| MTL-darts2nd (Maninis et al., 2019) | 2.75±0.06 | 3.3 | 4.2 |
| TETLO-darts2nd (ours) | **2.62**±0.06 | 3.2 | 4.0 |
| *Pdarts (Chen et al., 2019) | 2.55±0.06 | 3.5 | 0.3 |
| ATT-pdarts (He et al., 2016b) | 2.63±0.10 | 3.5 | 0.5 |
| ES-pdarts (Kokkinos, 2017) | 2.59±0.05 | 3.6 | 0.3 |
| MTL-pdarts (Maninis et al., 2019) | 2.57±0.07 | 3.4 | 0.4 |
| TETLO-pdarts (ours) | **2.47**±0.03 | 3.4 | 0.3 |
| *Pcdarts (Xu et al., 2020) | 2.57±0.07 | 3.6 | 0.1 |
| ATT-pcdarts (He et al., 2016b) | 2.68±0.06 | 3.5 | 0.2 |
| ES-pcdarts (Kokkinos, 2017) | 2.63±0.10 | 3.6 | 0.1 |
| MTL-pcdarts (Maninis et al., 2019) | 2.60±0.05 | 3.6 | 0.1 |
| TETLO-pcdarts (ours) | **2.48**±0.06 | 3.5 | 0.1 |
| *Prdarts (Zhou et al., 2020b) | 2.37±0.03 | 3.5 | 0.2 |
| ATT-prdarts (He et al., 2016b) | 2.45±0.07 | 3.5 | 0.4 |
| ES-prdarts (Kokkinos, 2017) | 2.60±0.14 | 3.6 | 0.2 |
| MTL-prdarts (Maninis et al., 2019) | 2.55±0.08 | 3.5 | 0.2 |
| TETLO-prdarts (ours) | **2.17**±0.02 | 3.4 | 0.2 |

**Second**, Table 2 shows our method performs better than MTL. In MTL, main and auxiliary tasks are learned simultaneously by minimizing the weighted sum of their training losses, which results in a competition between these two tasks: more decrease of auxiliary loss incurs less decrease of main loss; vice versa. In contrast, our method performs these two tasks sequentially. The auxiliary task helps the main task to learn instead of competing with it. **Third**, our method performs better than ATT. ATT is a classic way of performing transfer learning, which pretrains a model in an auxiliary task, then transfers this model to perform the main task. Our method performs transfer learning in a reverse direction: transfer is performed from main model to auxiliary model; quality of the main model is evaluated by its transferability to the auxiliary model. Results show our proposed direction of performing transfer is more effective. **Fourth**, our method outperforms ES. ES requires the main

model and auxiliary model to have the same encoder, which might be too restrictive and prohibit the learning of task-specific representations. Main and auxiliary tasks are different. A good representation beneficial for one task may not be suitable for the other. In contrast, our method allows different tasks to have different encoders, which facilitates task-specific representation learning. **Fifth**, while our TETLO framework can greatly help to improve the quality of searched architectures, it has similar model parameter number and search cost to baselines. The memory costs (Table 21) of our methods are also similar to those of baselines.

Table 3 show results on CIFAR-10. From this table, we make similar observations as those in Table 2. Our methods outperform vanilla DARTS, P-DARTS, PC-DARTS, PR-DARTS, and achieve lower errors than ATT, ES, MTL. The analysis of reasons is similar to that for results in Table 2.

## 4.5 IMAGENET RESULTS UNDER SEARCH SPACES OF DARTS, P-DARTS, PC-DARTS

For ImageNet, following (Xu et al., 2020), 10% of the 1.2M images are randomly sampled as a training set and another 2.5% are used for validation. Table 4 shows top-1 and top-5 test errors on ImageNet, where our methods outperform DARTS, P-DARTS, and PC-DARTS. These results again demonstrate the effectiveness of our method in improving the main encoder architecture by encouraging good transferability to an auxiliary model. Our method performs better than MTL, ATT, and ES, which further demonstrates the benefits of performing main and auxiliary tasks sequentially than jointly, transferring knowledge from the main model to an auxiliary instead of in the opposite direction, and learning task-specific representations.

## 4.6 ABLATION STUDIES

In this section, we perform ablation studies to understand the contributions of individual components in our framework.

**Ablation on knowledge transfer approaches.** We compare our quadruple relative similarity based knowledge transfer approach with: **1)** L2 regularization on embeddings (RegEmbed) (Romero et al., 2014): encouraging embeddings generated by auxiliary encoder to have small L2 distance with those generated by main encoder; **2)** pseudo-labeling (PL) (Zhang et al., 2017; Lan et al., 2018; Pham et al., 2020): the main model generates pseudo-labels, which are used to train the auxiliary; **3)** pairwise similarity (PS) (Chen et al., 2020a): the main model annotates whether two images are similar or dissimilar and the auxiliary fits these similarity labels; and **4)** triple-wise similarity (TS) (You et al., 2017): the main model compares similarity among three instances (e.g., $x$ is more similar to $y$ than to $z$) and the auxiliary fits these triple-wise similarities. Please refer to Appendix V for detailed setup of these baselines.

Table 4: Top-1 and top-5 classification errors on ImageNet test set. TETLO-darts2nd-cifar10: an architecture searched by applying TETLO to DARTS-2nd on CIFAR-10. Results marked with * are taken from DARTS$^-$ (Chu et al., 2020a), DrNAS (Chen et al., 2020b), OFA (Cai et al., 2020a), FB-Net (Wu et al., 2019), $\beta$-DARTS (Ye et al., 2022), and AG-NAS (Sun et al., 2022). The other notations are the same as those in Table 2.

| Method | Top-1 | Top-5 |
|---|---|---|
| *Inception-v1 (Szegedy et al., 2015) | 30.2 | 10.1 |
| *ShuffleNet 2× (v2) (Ma et al., 2018) | 25.1 | 7.6 |
| *NASNet-A (Zoph et al., 2018) | 26.0 | 8.4 |
| *AmoebaNet-C (Real et al., 2019) | 24.3 | 7.6 |
| *SDARTS-ADV-CIFAR10 (Chen & Hsieh, 2020) | 25.2 | 7.8 |
| *PC-DARTS-CIFAR10 (Xu et al., 2020) | 25.1 | 7.8 |
| *FBNet-C (Wu et al., 2019) | 25.1 | - |
| *ProxylessNAS-ImageNet (Cai et al., 2019) | 24.9 | 7.5 |
| *FairDARTS-ImageNet (Chu et al., 2019) | 24.4 | 7.4 |
| *PR-DARTS (Zhou et al., 2020b) | 24.1 | 7.3 |
| *DARTS$^+$-CIFAR100 (Liang et al., 2019) | 23.7 | 7.2 |
| *OFA large (Cai et al., 2020a) | 20.0 | - |
| *DOTS (Gu et al., 2021) | 24.3 | 7.4 |
| *$\beta$-DARTS (Ye et al., 2022) | 23.9 | 7.0 |
| *AGNAS (Sun et al., 2022) | 23.4 | 6.8 |
| *Darts2nd-cifar10 (Liu et al., 2019a) | 26.7 | 8.7 |
| ATT-darts2nd-cifar10 (He et al., 2016b) | 26.3 | 8.2 |
| ES-darts2nd-cifar10 (Kokkinos, 2017) | 26.4 | 8.5 |
| MTL-darts2nd-cifar10 (Maninis et al., 2019) | 26.4 | 8.3 |
| TETLO-darts2nd-cifar10 (ours) | **25.4** | **7.8** |
| *Pdarts-cifar10 (Chen et al., 2019) | 24.4 | 7.4 |
| ATT-pdarts-cifar10 (He et al., 2016b) | 24.4 | 7.2 |
| ES-pdarts-cifar10 (Kokkinos, 2017) | 24.4 | 7.3 |
| MTL-pdarts-cifar10 (Maninis et al., 2019) | 24.3 | 7.2 |
| TETLO-pdarts-cifar10 (ours) | **23.6** | **7.0** |
| *Pdarts-cifar100 (Chen et al., 2019) | 24.7 | 7.5 |
| ATT-pdarts-cifar100 (He et al., 2016b) | 24.6 | 7.5 |
| ES-pdarts-cifar100 (Kokkinos, 2017) | 24.6 | 7.5 |
| MTL-pdarts-cifar100 (Maninis et al., 2019) | 24.7 | 7.7 |
| TETLO-pdarts-cifar100 (ours) | **24.1** | **7.3** |
| *Pcdarts-ImageNet (Xu et al., 2020) | 24.2 | 7.3 |
| ATT-pcdarts-ImageNet (He et al., 2016b) | 24.0 | 7.1 |
| ES-pcdarts-ImageNet (Kokkinos, 2017) | 24.1 | 7.2 |
| MTL-pcdarts-ImageNet (Maninis et al., 2019) | 23.9 | 7.3 |
| TETLO-pcdarts-ImageNet (ours) | **23.3** | **6.2** |

two images are similar or dissimilar and the auxiliary fits these similarity labels; and **4)** triple-wise similarity (TS) (You et al., 2017): the main model compares similarity among three instances (e.g., $x$ is more similar to $y$ than to $z$) and the auxiliary fits these triple-wise similarities. Please refer to Appendix V for detailed setup of these baselines.

Figure 4 shows the results. We observe the following. **First**, our method works better than RegEmbed and PL. These two baselines transfer knowledge on individual data instances without considering the relationship between instances. In contrast, our method considers relative similarity relationships among four data instances during knowledge transfer. **Second**, our method works better than

PS and TS. PS and TS are limited to capturing second-order and third-order relationships among instances while our method can capture fourth-order relationships.

Figure 6 shows 4-nearest neighbors retrieved by different methods, for some randomly sampled CIRAR-100 test images. We use the trained main encoder to generate image encodings and calculate cosine similarity on these

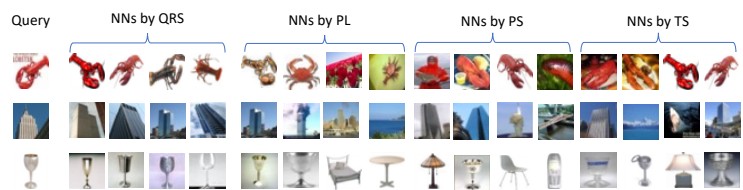

Figure 3: Nearest neighbors retrieved by different methods.

encodings. Nearest neighbors are retrieved based on cosine similarity. Compared with baselines, nearest neighbors retrieved by QRS are more semantically similar to query images. By capturing 4th-order similarity relationships among instances, QRS can more effectively group similar images together, which is good for classification.

**Ablation on removing main validation loss.** Next, we perform another ablation study – "auxiliary validation only": main encoder architecture is updated by minimizing auxiliary validation loss only, without considering main validation loss. TETLO is applied to DARTS-2nd. Table 5 shows the results. On both datasets, minimizing the sum of main and auxiliary validation losses ("main+auxiliary") results in better architectures than minimizing auxiliary validation loss only. The reason is that auxiliary validation loss indirectly measures the quality of architectures. In contrast, the main validation loss directly depends on

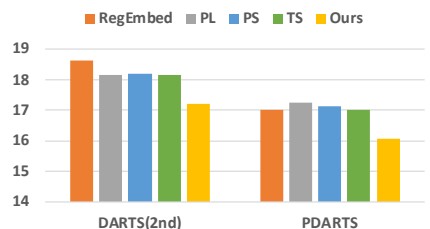

Figure 4: Comparison of knowledge transfer approaches. The y-axis is test error on CIFAR-100.

main encoder architectures and can serve as a direct (hence stronger) signal to guide architecture search. In the end, combining the direct signal (main validation loss) and indirect signal (auxiliary validation loss) together is more beneficial than using the indirect signal only.

**Ablation on tradeoff parameters.** Finally, we investigate how performance changes with tradeoff parameters $\gamma$ and $\lambda$ in Eq.(3). TETLO is applied to DARTS-2nd. Figure 5 shows the results on a 5K held-out validation set of CIFAR-100. A value of $\gamma$ in the middle ground that properly balances validation losses of the main and auxiliary models yields the optimal performance. This is also the case for $\lambda$.

Table 5: Test errors in the ablation of "auxiliary validation only". C denotes CIFAR.

| Method | Error (%) |
|---|---|
| Auxiliary only (C100) | 19.54±0.13 |
| Main + auxiliary (C100) | **17.21**±0.10 |
| Auxiliary only (C10) | 2.88±0.09 |
| Main + auxiliary (C10) | **2.62**±0.06 |

**Other ablations.** We conduct other ablations, such as the following one: ignoring the auxiliary task entirely, defining a QRS loss on the validation data, and using this QRS loss together with the classification loss on the validation set to optimize architectures. Please refer to Appendix N for details.

## 5 CONCLUSIONS

We propose a transferability-encouraging tri-level optimization framework which aims to improve the generalizability and stability of differentiable neural architecture search methods by encouraging good transferability to auxiliary

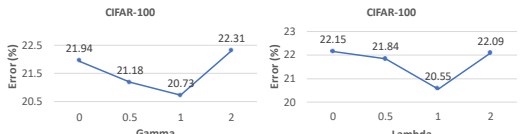

Figure 5: How errors change with $\gamma$ (left) and $\lambda$ (right).

models. Our framework performs three learning stages end-to-end. First, we train a main model. Second, the knowledge of the main model is transferred to an auxiliary model based on matching quadruple relative similarities. Third, the main-to-auxiliary transferability is measured and the main model is improved by maximizing this transferability. Experiments on a variety of datasets demonstrate the efficacy of our methods.

REPRODUCIBILITY STATEMENT

Datasets used in experiments are publicly available and are cited and described in Section 4.1. Major training details including hyperparameters, parameter tuning strategies, data splits, etc. are specified in Section 4.2, Section M, Section M, Section O, Section P, Section Q, Section T, and Section V. Standard deviations of results (after running experiments multiple times with random seeds) are reported in Table 2, Table 5, Table 16, Table 3, and Table 22. Statistical significance tests are reported in Section R. Computation time is reported in Table 2, Table 3, Table 19, Table 20, and Table 25. GPU type is reported in Section M.1.

ACKNOWLEDGEMENTS

We thank all the reviewers for invaluable comments and feedback.

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

Table 6: Test error (%) on CIFAR-10, when validation data has label noise and loss function is cross-entropy.

| Method | Error(%) |
|---|---|
| Darts2nd (Liu et al., 2019a) | 21.35±0.40 |
| TETLO-darts2nd (ours) | **18.71**±0.33 |

Table 7: Test error (%) on CIFAR-10, when validation dataset is class-imbalanced.

| Method | Error(%) |
|---|---|
| Darts2nd (Liu et al., 2019a) | 16.02±0.25 |
| TETLO-darts2nd (ours) | **14.29**±0.17 |

## APPENDIX

## A   LABEL NOISE AND CLASS IMBALANCE IN VALIDATION DATASET.

In this section, we discuss the case where validation data has label noise or class imbalance. In any differentiable neural architecture search method, there is always a validation set used to update architecture variables. This validation set is used as the auxiliary model's validation set in our method. As long as a differentiable NAS method has access to a class-balanced and non-noisy validation set, the auxiliary model in our method has access to such a validation set as well. In cases where the validation dataset has issues of class imbalance and label noise, there have been NAS methods (Chen et al., 2020c; Timofeev et al., 2021) developed to cope with these issues. Our framework can be combined with these NAS methods to learn high-performance architectures from class-imbalanced and noisy validation data.

We performed experiments to investigate how our method performs when the validation dataset has label noises. Following (Chen et al., 2020c; Kumar & Sastry, 2018), we added symmetric label noise (with parameter $\eta = 0.6$) to validation data: the probability that one class is corrupted to be another class is equal. Training and test sets are clean. The experiments were conducted on CIFAR-10. Training and validation losses are cross-entropy. All methods perform architecture search using the same noisy validation dataset. Table 6 shows test errors (%) on CIFAR-10. As can be seen, our TETLO-darts2nd outperforms vanilla Darts2nd when validation data has label noises.

To combat label noise in validation dataset, we can combine our approach with robust NAS methods such as (Chen et al., 2020c) which uses robust log losses (Kumar & Sastry, 2018) as training and validation losses. Table 6 shows the results (both TETLO-darts2nd and Darts2nd use robust log loss). As can be seen, our TETLO-darts2nd achieves better performance than vanilla Darts2nd. Compared with Table 6, we can see that using robust log loss achieves better performance than using cross-entropy loss.

We also performed experiments to investigate how our method performs when the validation dataset has class imbalance. The experiment was conducted on CIFAR-10. In the original training, validation, and test set of CIFAR-10, the classes are balanced. Following (Timofeev et al., 2021), we transformed the validation set of CIFAR-10 into an imbalanced dataset by reducing the number of data examples for each class according to an exponential function. Training and test sets are balanced. All methods perform architecture search using the same class-imbalanced validation dataset. Table 6 shows test errors (%). As can be seen, our TETLO-darts2nd outperforms vanilla Darts2nd when validation data has class imbalance.

To deal with class-imbalance in validation dataset, we can combine our approach with a class reweighting method proposed by (Cui et al., 2019). Table 8 shows the results (both TETLO-darts2nd and Darts2nd use this class reweighting method). As can be seen, our TETLO-darts2nd achieves better performance than vanilla Darts2nd. Compared with Table 7, we can see that using class reweighting achieves better performance than not using class reweighting.

Table 8: Test error (%) on CIFAR-10, when validation dataset is class-imbalanced and class reweighting is used.

| Method | Error(%) |
|---|---|
| Darts2nd (Liu et al., 2019a) | 13.51±0.29 |
| TETLO-darts2nd (ours) | **11.74**±0.13 |

Table 9: Top-1 error (%) on ImageNet.

| Method | Top-1 error(%) |
|---|---|
| OFA w/o PS | 27.6 |
| OFA w/ PS | 24.0 |
| OFA w/ PS #25 | 23.6 |
| OFA w/ PS #75 | 23.1 |
| Ours-darts2nd-cifar10 | 25.4 |
| Ours-pdarts-cifar10 | 23.6 |
| Ours-pdarts-cifar100 | 24.1 |
| Ours-pcdarts-ImageNet | 23.3 |

## B  COMPARE OUR METHOD WITH OFA

OFA-large achieves the best performance because it trains many sub-architectures progressively, from the largest sub-architecture to gradually smaller sub-architectures. It takes 1200 GPU hours to train so many sub-architectures. In contrast, our method trains a single searched sub-architecture which takes much less time. Table 9 compares our method with other variants of OFA. As can be seen, some variants of our method outperforms some variants of OFA.

Our method and OFA both perform knowledge transfer (from model A to model B). The difference is: in our method, model B (the auxiliary model) is leveraged to improve model A (the main model) by minimizing B's validation loss. In contrast, in OFA, model A is leveraged to improve model B.

## C  SEARCH COSTS UNDER DIFFERENT KNOWLEDGE TRANSFER APPROACHES.

Table 10 shows the search costs (GPU days) under different knowledge transfer (KT) approaches including QRS, pairwise similarity (PS, second order), and triple-wise similarity (TS, third order). The experiments were conducted on CIFAR-100. As the number of orders increases (from 2 in PS, to 3 in TS, to 4 in QRS), the search costs increase. But the increase is not substantial. As shown in Figure 4 in the main paper, QRS achieves much better classification performance than TS and PS. Overall, the gain in classification performance achieved by QRS over TS and PS outweighs the slightly increased search cost.

## D  ABLATION ON AUXILIARY MODELS.

We study how the performance of the main model is affected by different auxiliary models and experimented with ResNet-18 (RN18), ResNet-50 (RN50), ResNet-101 (RN101), and a simple CNN proposed in (Vinyals et al., 2016), on DARTS, PCDARTS, and PDARTS. Table 11 shows the results. RN50 works better than RN18. RN50 has more layers and is more expressive than RN18. This demonstrates that a more expressive auxiliary helps to train a stronger main model. Intuitively, the predictive performance of a more expressive auxiliary is more sensitive to knowledge generated by the main model; consequently, minimizing the auxiliary's validation loss can more significantly change the fidelity of knowledge and further the predictive power of the main model. We provide empirical evidence in Figure 6, by measuring the correlation between classification errors of auxiliaries and errors of QRS relationships. RN50's correlation is higher than RN18's. Other observations include: 1) ResNet variants work better than the simple CNN; 2) ResNet-50 is on par with ResNet-101.

Table 10: Search costs (GPU days) under different knowledge transfer approaches.

| Search Method | KT Method | Search Cost |
|---|---|---|
| Ours-Darts2nd | PS | 3.8 |
| | TS | 3.9 |
| | QRS | 4.0 |

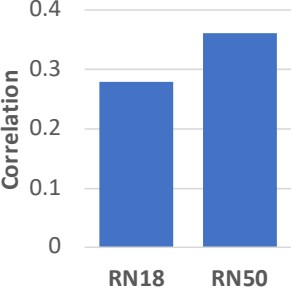

Figure 6: Correlation between classification errors of auxiliaries and errors of QRS relationships.

## E    APPLY TETLO TO DARTS WITH FIRST ORDER.

We apply our TETLO method to DARTS with first order (DARTS-1st) as well. Table 12 and 13 show results on CIFAR-100 and CIFAR-10. Our method outperforms vanilla DARTS-1st.

## F    ARCHITECTURE TRANSFERABILITY BETWEEN CIFAR-10 AND CIFAR-100.

Table 14 shows the results of architecture transferability between CIFAR-10 and CIFAR-100. Cifar10→Cifar100 denotes that the architecture is trained on CIFAR-10 and evaluated on CIFAR-100. Cifar100→Cifar10 denotes that the architecture is trained on CIFAR-100 and evaluated on CIFAR-10. As can be seen, our method has better architecture transferability between CIFAR-10 and CIFAR-100 than vanilla DARTS2nd and PDARTS, which further demonstrates the effectiveness of our transferability-encouraging three-level optimization framework.

## G    MEASURE STABILITY

Following (Chen & Hsieh, 2020), we measure the stability of differentiable NAS methods on the NAS-Bench-1Shot1 (Zela et al., 2020) benchmark. NAS-Bench-1Shot1 has three search spaces for CIFAR-100. Following (Chen & Hsieh, 2020), we perform architecture search for 100 epochs and measure the architectures searched at different epochs on a test set. If the average test accuracy for epoch 90-100 is larger than that for epoch 45-55, then the architecture is considered to be degenerate. For each NAS method, we run it for 10 times and measure how many times the searched architecture is degenerate. Table 15 shows the results. As can be seen, our TETLO-darts2nd has fewer degenerate times than Darts2nd, which demonstrates that our method is more stable than vanilla Darts2nd, thanks to its transferability-encouraging mechanism.

Table 11: Ablation on auxiliaries.

| Method | Error (%) |
|---|---|
| SCNN-darts2nd | 18.06±0.11 |
| RN18-darts2nd | 17.63±0.09 |
| RN50-darts2nd | **17.15**±0.14 |
| RN101-darts2nd | 17.32±0.08 |
| SCNN-pdarts | 17.11±0.07 |
| RN18-pdarts | 16.24±0.09 |
| RN50-pdarts | 16.02±0.06 |
| RN101-pdarts | **15.97**±0.10 |

Table 12: Results of TETLO-darts1st on CIFAR-100, including classification error (%) on the test set, number of model weights (millions), and search cost (GPU days).

| Method | Error(%) | Param(M) | Cost |
|---|---|---|---|
| Darts1st (Liu et al., 2019a) | 20.49±0.27 | 3.4 | 1.5 |
| TETLO-darts1st (ours) | **18.46**±0.13 | 3.3 | 1.5 |

Table 13: Results of TETLO-darts1st on CIFAR-10, including classification error (%) on the test set, number of model weights (millions), and search cost (GPU days).

| Method | Error(%) | Param(M) | Cost |
|---|---|---|---|
| Darts1st (Liu et al., 2019a) | 3.00±0.14 | 3.3 | 1.5 |
| TETLO-darts1st (ours) | **2.80**±0.09 | 3.1 | 1.5 |

Table 14: Architecture transferability between CIFAR-10 and CIFAR-100. The results are classification errors (%) on test sets.

| Method | Cifar10→Cifar100 | Cifar100→Cifar10 |
|---|---|---|
| Darts2nd (Liu et al., 2019a) | 21.42±0.26 | 2.82±0.07 |
| TETLO-darts2nd (ours) | **18.05**±0.14 | **2.69**±0.05 |
| Pdarts (Chen et al., 2019) | 18.65±0.10 | 2.67±0.09 |
| TETLO-pdarts (ours) | **16.82**±0.12 | **2.53**±0.04 |

Table 15: Times of architecture degeneration.

| Method | Search space 1 | Search space 2 | Search space 3 |
|---|---|---|---|
| Darts2nd (Liu et al., 2019a) | 8 | 5 | 9 |
| TETLO-darts2nd (ours) | **2** | **1** | **2** |

# H   OPTIMIZATION ALGORITHM

We use a well-established algorithm developed in (Liu et al., 2019a) to solve the proposed TETLO problem. Theoretic convergence of this algorithm has been broadly analyzed in (Ghadimi & Wang, 2018; Grazzi et al., 2020; Ji et al., 2021; Liu et al., 2021; Yang et al., 2021). At each level of optimization problem, the optimal solution (on the left-hand side of the equal sign, marked with *), its exact value is computationally expensive to compute. To address this problem, following (Liu et al., 2019a), we approximate the optimal solution using a one-step gradient descent update and plug the approximation into the next level of optimization problem. In the sequel, $\frac{\partial \cdot}{\partial \cdot}$ denotes partial derivative. $\frac{d \cdot}{d \cdot}$ denotes an ordinary derivative.

With architecture tying, $A$ is the same as $M$. The optimization problem becomes:

Following (Liu et al., 2019a), we approximate $E_m^*(M)$ using

$$E_m' = E_m - \xi_m \frac{dL_m(M, E_m, H_m, D_m^{(tr)})}{dE_m}. \tag{5}$$

We approximate $H_m^*(M)$ using

$$H_m' = H_m - \xi_m \frac{dL_m(M, E_m, H_m, D_m^{(tr)})}{dH_m}, \tag{6}$$

where $\xi_m$ is a learning rate. Substituting $E_m'$ into the loss function at the second stage, we obtain an approximated loss:

$$O = L_a(A, E_a, H_a, D_a^{(tr)}) + \lambda \sum_{x,y \in D_a^{(tr)}} \max(0, -(c(x,y;E_m',M) - c(z,w;E_m',M))(c(x,y;E_a,A) - c(z,w;E_a,A))). \tag{7}$$

Next, we approximate $E_a^*(M,A)$ and $H_a^*(A)$ using one-step gradient descent update of $E_a$ and $H_a$ w.r.t $O$:

$$E_a' = E_a - \xi_a \frac{dO}{dE_a}, \tag{8}$$

$$H_a' = H_a - \xi_a \frac{dO}{dH_a}. \tag{9}$$

Finally, we plug these approximations into the validation loss at the third stage and get an approximated validation loss $P$:

$$P = L_m(M, E_m', H_m', D_m^{(val)}) + \gamma L_a(A, E_a', H_a', D_a^{(val)}). \tag{10}$$

We can update the meta parameters $M$ and $A$ by descending the gradient of $P$ w.r.t $M$ and $A$:

$$M \leftarrow M - \eta \frac{dP}{dM}, \ A \leftarrow A - \eta \frac{dP}{dA}, \tag{11}$$

where

$$\frac{dP}{dM} = \frac{dL_m(M, E_m', H_m', D_m^{(val)})}{dM} + \gamma \frac{dL_a(A, E_a', H_a', D_a^{(val)})}{dM}, \tag{12}$$

and

$$\frac{dP}{dA} = \gamma \frac{dL_a(A, E_a', H_a', D_a^{(val)})}{dA}. \tag{13}$$

$\frac{dL_m(M, E_m', H_m', D_m^{(val)})}{dM}$ can be calculated as:

$$\frac{dL_m(M, E_m', H_m', D_m^{(val)})}{dM} = \frac{\partial L_m(M, E_m', H_m', D_m^{(val)})}{\partial M} + \frac{dE_m'}{dM} \frac{\partial L_m(M, E_m', H_m', D_m^{(val)})}{\partial E_m'} + \frac{dH_m'}{dM} \frac{\partial L_m(M, E_m', H_m', D_m^{(val)})}{\partial H_m'} \tag{14}$$

$\frac{dL_a(A, E_a', H_a', D_a^{(val)})}{dM}$ can be calculated as:

$$\frac{dL_a(A, E_a', H_a', D_a^{(val)})}{dM} = \frac{dE_a'}{dM} \frac{dL_a(A, E_a', H_a', D_a^{(val)})}{\partial E_a'}, \tag{15}$$

where

$$\frac{dE'_a}{dM} = \frac{\partial E'_a}{\partial M} + \frac{dE'_m}{dM}\frac{\partial E'_a}{\partial E'_m} = -\xi_a\left(\frac{\partial \nabla_{E_a}O}{\partial M} + \frac{dE'_m}{dM}\frac{\partial \nabla_{E_a}O}{\partial E'_m}\right) \tag{16}$$

In Eq.(13), $\frac{dL_a(A,E'_a,H'_a,D_a^{(val)})}{dA}$ can be calculated as:

$$\frac{dL_a(A,E'_a,H'_a,D_a^{(val)})}{dA} = \frac{\partial L_a(A,E'_a,H'_a,D_a^{(val)})}{\partial A} + \frac{dE'_a}{dA}\frac{\partial L_a(A,E'_a,H'_a,D_a^{(val)})}{\partial E'_a} + \frac{dH'_a}{dA}\frac{\partial L_a(A,E'_a,H'_a,D_a^{(val)})}{\partial H'_a}$$
$$\tag{17}$$

In the above equations, we have:

$$\frac{dE'_m}{dM} = -\xi_m \nabla^2_{M,E_m}L_m(M,E_m,H_m,D_m^{(tr)}) \tag{18}$$

$$\frac{dH'_m}{dM} = -\xi_m \nabla^2_{M,H_m}L_m(M,E_m,H_m,D_m^{(tr)}) \tag{19}$$

$$\frac{dE'_a}{dA} = -\xi_a \nabla^2_{A,E_a}L_a(A,E_a,H_a,D_a^{(tr)})$$
$$-\xi_a\lambda\nabla^2_{A,E_a}\sum_{x,y\in D_a^{(tr)}}\max(0,-(c(x,y;E'_m,M)-c(z,w;E'_m,M))(c(x,y;E_a,A)-c(z,w;E_a,A)))$$
$$\tag{20}$$

$$\frac{dH'_a}{dA} = -\xi_a \nabla^2_{A,H_a}L_a(A,E_a,H_a,D_a^{(tr)}) \tag{21}$$

Similar to (Liu et al., 2019a), we approximate matrix-vector multiplication terms using finite differences.

The gradient descent update of $M$ and $A$ in equation 11 can run one or more steps. After $M$ and $A$ are updated, the one-step gradient-descent approximations (in Equation 5,6,8,9), which are functions of $M$ and $A$, need to be re-updated. Then, the gradients of $M$ and $A$, which are functions of one-step gradient-descent approximations, need to be re-calculated and are used to refresh $M$ and $A$. In sum, the updates of $M/A$ and the updates of one-step gradient-descent approximations mutually depend on each other. These updates are performed iteratively until convergence. Algorithm 1 shows the algorithm.

**while** *not converged* **do**
    1. Update the approximation $E'_m$ of $E^*_m(M)$ using Eq.(5)
    2. Update the approximation $H'_m$ of $H^*_m(M)$ using Eq.(6)
    3. Update the approximation $E'_a$ of $E^*_a(M,A)$ using Eq.(8)
    4. Update the approximation $H'_a$ of $H^*_a(A)$ using Eq.(9)
    5. Update architectures $M$ and $A$ using Eq.(11)
**end**

**Algorithm 1:** Optimization algorithm for TETLO

Though using an approximated optimization algorithm, our framework is still a valid tri-level framework. As shown in equations 15, 16, 18, the gradient of the validation loss w.r.t to $M$ involves the product of three gradients: gradient of the validation loss w.r.t to $E'_a$ , gradient of $E'_a$ w.r.t $E'_m$ , gradient of $E'_m$ w.r.t $M$. The gradient information is propagated from stage 3 to stage 2, then to stage 1. This shows that the tri-level optimization formulation of our framework is well preserved in the approximated algorithm.

## H.1 NUMBER OF GRADIENT DESCENT STEPS

In the above algorithm, at each level, we use one-step gradient descent (GD) to approximate the optimal solution. We are interested in investigating whether more ($> 1$) GD steps can result in better performance. We experimented with more GD steps at each level. The results are in Table 16. As can be seen, more GD steps do not differ significantly from one-step GD. This indicates that one-step gradient update yields a sufficiently good approximation to the optimal solution.

| Method | Dataset | Num. of GD steps | Test error (%) |
|---|---|---|---|
| TETLO-darts2nd (ours) | CIFAR100 | 1 | 17.21±0.10 |
| TETLO-darts2nd (ours) | CIFAR100 | 2 | 17.16±0.13 |
| TETLO-darts2nd (ours) | CIFAR100 | 3 | 17.11±0.17 |
| TETLO-darts2nd (ours) | CIFAR10 | 1 | 2.62±0.06 |
| TETLO-darts2nd (ours) | CIFAR10 | 2 | 2.59±0.07 |
| TETLO-darts2nd (ours) | CIFAR10 | 3 | 2.57±0.06 |
| TETLO-pdarts (ours) | CIFAR100 | 1 | 16.05±0.08 |
| TETLO-pdarts (ours) | CIFAR100 | 2 | 16.01±0.11 |
| TETLO-pdarts (ours) | CIFAR100 | 3 | 15.98±0.07 |
| TETLO-pdarts (ours) | CIFAR10 | 1 | 2.47±0.03 |
| TETLO-pdarts (ours) | CIFAR10 | 2 | 2.47±0.04 |
| TETLO-pdarts (ours) | CIFAR10 | 3 | 2.46±0.04 |

Table 16: Test errors under different numbers of GD steps.

Table 17: Top-1 and top-5 classification errors on ImageNet test set, number of parameters (millions), and search costs (GPU days).

| Method | Top-1 | Top-5 | #Parameters (M) | Search Costs (GPU days) |
|---|---|---|---|---|
| FBNet-A | 27.0 | - | 4.3 | 9 |
| FBNet-B | 25.9 | - | 4.5 | 9 |
| FBNet-C | 25.2 | - | 5.5 | 9 |
| OFA w/o PS | 27.6 | - | - | 1.7 |
| OFA w/ PS | 24.0 | - | - | 1.7 |
| OFA w/ PS #25 | 23.6 | - | - | 1.7 |
| OFA w/ PS #75 | 23.1 | - | - | 1.7 |
| OFA-large | 20.0 | - | - | 1.7 |
| Ours-darts2nd-cifar10 | 25.4 | 7.8 | 4.6 | 4.0 |
| Ours-pdarts-cifar10 | 23.6 | 7.0 | 4.8 | 0.3 |
| Ours-pdarts-cifar100 | 24.1 | 7.3 | 5.1 | 0.3 |
| Ours-pcdarts-ImageNet | 23.3 | 6.2 | 5.3 | 4.0 |

## I COMPARE OUR METHOD WITH FBNET AND ONCE-FOR-ALL (OFA) NETWORKS

Table 17 compares our method with FBNet (Wu et al., 2019) and Once-for-All (OFA) (Cai et al., 2020b) networks on ImageNet. The classification performance and search costs of our methods are on par with those of OFA in general. The search costs of our methods are lower than those of FBNet while our methods achieve classification performance similar to that of FBNet.

## J NUMBER OF SKIP CONNECTIONS AND CONVOLUTIONS

Table 18 shows the number of skip connections and convolutions in searched architectures. Darts-Cifar10-Normal denotes a normal cell searched on Cifar10 by Darts. Reduction denotes a reduction cell. The search space is that in DARTS. As can be seen, in most cases, the number of skip connections in cells searched by our methods is smaller than that in baselines, which demonstrates the effectiveness of our methods in avoiding an excessive number of skip connections which causes performance degradation.

## K SEARCH COSTS FOR ROBUSTNESS EXPERIMENTS ON S1-S4

Table 19 and 20 shows search costs for robustness experiments on S1-S4 and CIFAR-10/100.

Table 18: Number of skip connections, convolutions, and poolings

| Cell | # skip connections | # 3x3 convolutions | # 5x5 convolutions | # poolings |
|---|---|---|---|---|
| Darts-Cifar10-Normal | 3 | 4 | 1 | 0 |
| Ours-Darts-Cifar10-Normal | 2 | 4 | 2 | 0 |
| Darts-Cifar10-Reduction | 3 | 3 | 0 | 2 |
| Ours-Darts-Cifar10-Reduction | 1 | 2 | 1 | 4 |
| Darts-Cifar100-Normal | 4 | 3 | 1 | 0 |
| Ours-Darts-Cifar100-Normal | 2 | 6 | 0 | 0 |
| Darts-Cifar100-Reduction | 3 | 2 | 0 | 3 |
| Ours-Darts-Cifar100-Reduction | 1 | 1 | 0 | 6 |
| Pcdarts-Cifar10-Normal | 3 | 3 | 1 | 1 |
| Ours-Pcdarts-Cifar10-Normal | 2 | 5 | 1 | 0 |
| Pcdarts-Cifar10-Reduction | 2 | 4 | 0 | 2 |
| Ours-Pcdarts-Cifar10-Reduction | 0 | 3 | 5 | 0 |
| Pcdarts-Cifar100-Normal | 1 | 4 | 1 | 2 |
| Ours-Pcdarts-Cifar100-Normal | 1 | 2 | 5 | 0 |
| Pcdarts-Cifar100-Reduction | 0 | 3 | 0 | 5 |
| Ours-Pcdarts-Cifar100-Reduction | 0 | 6 | 2 | 0 |
| Pdarts-Cifar10-Normal | 2 | 2 | 2 | 2 |
| Ours-Pdarts-Cifar10-Normal | 2 | 5 | 1 | 0 |
| Pdarts-Cifar10-Reduction | 2 | 2 | 0 | 4 |
| Ours-Pdarts-Cifar10-Reduction | 0 | 0 | 5 | 3 |
| Pdarts-Cifar100-Normal | 1 | 5 | 0 | 2 |
| Ours-Pdarts-Cifar100-Normal | 1 | 3 | 3 | 1 |
| Pdarts-Cifar100-Reduction | 1 | 3 | 0 | 4 |
| Ours-Pdarts-Cifar100-Reduction | 0 | 8 | 0 | 0 |
| Prdarts-Cifar10-Normal | 2 | 5 | 0 | 1 |
| Ours-Prdarts-Cifar10-Normal | 2 | 6 | 0 | 0 |
| Prdarts-Cifar10-Reduction | 2 | 1 | 2 | 3 |
| Ours-Prdarts-Cifar10-Reduction | 0 | 3 | 2 | 3 |
| Prdarts-Cifar100-Normal | 3 | 3 | 1 | 1 |
| Ours-Prdarts-Cifar100-Normal | 1 | 4 | 3 | 0 |
| Prdarts-Cifar100-Reduction | 3 | 3 | 0 | 2 |
| Ours-Prdarts-Cifar100-Reduction | 0 | 3 | 2 | 3 |

Table 19: Search costs for S1-S4 on CIFAR-10. Search cost is measured by GPU hours on an Nvidia 1080Ti.

| Method | S1 | S2 | S3 | S4 |
|---|---|---|---|---|
| DARTS | 6.1 | 8.9 | 10.4 | 8.7 |
| RDARTS-L2 | 6.2 | 9.1 | 10.4 | 8.8 |
| DARTS-ES | 10.1 | 18.3 | 15.1 | 16.7 |
| DARTS- | 6.0 | 8.9 | 10.5 | 8.9 |
| SDARTS-ADV | 7.8 | 11.4 | 13.7 | 11.4 |
| MTL | 6.3 | 8.9 | 10.6 | 8.8 |
| Ours | 6.1 | 8.7 | 10.3 | 8.7 |

Table 20: Search costs for S1-S4 on CIFAR-100. Search cost is measured by GPU hours on an Nvidia 1080Ti.

| Method | S1 | S2 | S3 | S4 |
|---|---|---|---|---|
| DARTS | 6.3 | 9.0 | 10.6 | 9.0 |
| RDARTS-L2 | 6.0 | 9.3 | 10.7 | 8.6 |
| DARTS-ES | 10.4 | 17.9 | 14.8 | 16.5 |
| DARTS- | 6.1 | 9.0 | 10.7 | 9.1 |
| SDARTS-ADV | 7.8 | 11.6 | 13.8 | 11.3 |
| MTL | 6.3 | 9.2 | 10.5 | 9.1 |
| Ours | 6.2 | 9.0 | 10.5 | 8.8 |

## L   MEMORY COSTS

Table 21 compares memory costs of different methods. As can be seen, the memory costs of our methods are similar to those of baselines while our methods achieve lower test errors than baselines.

| Method | Error-Cifar100 | Error-Cifar10 | Memory cost on Cifar100 | Memory cost on Cifar10 |
|---|---|---|---|---|
| Darts2nd | 20.58±0.44 | 2.76±0.09 | 11053 | 11008 |
| TETLO-darts2nd (ours) | **17.21**±0.10 | **2.62**±0.06 | 11091 | 11049 |
| Pdarts | 17.42±0.14 | 2.55±0.06 | 9659 | 9721 |
| TETLO-pcdarts (ours) | **16.05**±0.08 | **2.47**±0.03 | 9692 | 9745 |
| Pcdarts | 17.96±0.15 | 2.57±0.07 | 10058 | 10024 |
| TETLO-pcdarts (ours) | **16.42**±0.07 | **2.48**±0.06 | 10083 | 10066 |
| Prdarts | 16.48±0.06 | 2.37±0.03 | 10159 | 10119 |
| TETLO-pcdarts (ours) | **15.81**±0.06 | **2.17**±0.02 | 10192 | 10176 |

Table 21: Test errors (%) memory costs on CIFAR-100 and CIFAR-10.

## M   EXPERIMENTAL DETAILS OF NEURAL ARCHITECTURE SEARCH

### M.1   ADDITIONAL COMMON SETTINGS

In each iteration, we randomly sample a small minibatch (300) of pairs to calculate quadruple relative similarity relations, instead of using all pairs. Calculation on the small minimatch is very efficient. Following (Liu et al., 2019a), we select the checkpoint at the final epoch to evaluate the test performance. No early stopping is involved. All GPU costs are measured on 1080Ti GPUs.

### M.2   DARTS BASED EXPERIMENTS

For methods based on DARTS2nd, including TETLO-darts2nd (ours), MTL-darts2nd, ES-darts2nd, ATT-darts2nd, the experimental settings are similar. In search spaces of DARTS, the candidate operations include: $3 \times 3$ and $5 \times 5$ separable convolutions, $3 \times 3$ and $5 \times 5$ dilated separable convolutions, $3 \times 3$ max pooling, $3 \times 3$ average pooling, identity, and zero. The network is a stack of multiple cells, each consisting of 7 nodes. The stride of all operations is set to 1. The convolved feature maps are padded to preserve their spatial resolution. The order for convolutional operations is ReLU-Conv-BN. Each separable convolution is applied twice. The convolutional cell has 7 nodes. The output node is the depthwise concatenation of all intermediate nodes, excluding the input nodes. We create a network by stacking 8 cells. The first and second nodes of cell $k$ are equal to the outputs of cell $k-2$ and cell $k-1$, respectively. 1x1 convolutions are inserted when necessary. Reduction cells are located at the 1/3 and 2/3 of the total depth of the network. In reduction cells, operations adjacent to the input nodes have a stride of 2.

For CIFAR-10 and CIFAR-100, during architecture search, the main network model is a stack of 8 cells, each consisting of 7 nodes, with the initial channel number set to 16. The search algorithm ran for 30 epochs with a batch size of 32. Network weights are optimized using SGD, with an initial learning rate of 0.025 (adjusted using a cosine decay scheduler), a momentum of 0.9, and a weight decay of 3e-4. The architecture variables were optimized using Adam (Kingma & Ba, 2014) with a learning rate of 0.001, a momentum of $(0.5, 0.999)$, and a weight decay of 0.001. The learning rate was scheduled with cosine scheduling. The architecture variables were initialized with zero initialization.

During architecture evaluation, for CIFAR-10 and CIFAR-100, a larger network of the main model is formed by stacking 20 copies of the searched cell. The composed large network is trained on the combination of $D_t^{(\mathrm{tr})}$ and $D_t^{(\mathrm{val})}$. The initial channel number was set to 36. We trained the network with a batch size of 96, an epoch number of 600. The SGD optimizer is used for weights training, with an initial learning rate of 0.025, a cosine decay scheduler, a batch size of 96, a momentum of 0.9, and a weight decay of 3e-4. On ImageNet, we evaluate two types of architectures: 1) those searched on a subset of ImageNet; 2) those searched on CIFAR-10 or CIFAR-100. In either type, 14 copies of optimally searched cells are stacked into a large network, which was trained on the

1.2M training images, with a batch size of 1024, an epoch number of 250, an initial learning rate of 0.5, and a weight decay of 3e-5. Initial channel number was set to 48. Cutout, path dropout of probability 0.2 and auxiliary towers with weight 0.4 were applied.

### M.3 PC-DARTS BASED EXPERIMENTS

For methods based on PC-DARTS, including TETLO-pcdarts (ours), MTL-pcdarts, ES-pcdarts, ATT-pcdarts, the experimental settings are similar. The search space of PC-DARTS follows that of DARTS. For architecture search on CIFAR-100 and CIFAR-10, the hyperparameter $K$ was set to 4. The network is a stack of 8 cells. Each cell contains 6 nodes. Initial channel number is set to 16. The architecture variables are trained using the Adam optimizer for 50 epochs. The learning rate is set to $6e-4$, without decay. The weight decay is set to $1e-3$. The momentum is set to $(0.5, 0.999)$. The network weight parameters are trained using SGD for 50 epochs. The initial learning rate is set to 0.1. Cosine scheduling is used to decay the learning rate, down to 0 without restart. The momentum is set to 0.9. The weight decay is set to $3e-4$. The batch size is set to 256. Warm-up is utilized: in the first 15 epochs, architecture variables are frozen and only network weights are optimized.

The settings for architecture evaluation on CIFAR-100 and CIFAR-10 follow those of DARTS. 18 normal cells and 2 reduction cells are stacked into a large network. The initial channel number is set to 36. The stacked network is trained from scratch using SGD for 600 epochs, with batch size 128, initial learning rate 0.025, momentum 0.9, weight decay $3e-4$, norm gradient clipping 5, drop-path rate 0.3, and cutout. The learning rate is decayed to 0 using cosine scheduling without restart.

We combine our method and PC-DARTS to directly search for architectures on ImageNet. The stacked network starts with three convolution layers which reduce the input image resolution from $224\times224$ to $28\times28$, using stride 2. After the three convolution layers, 6 normal cells and 2 reduction cells are stacked. Each cell consists of $N = 6$ nodes. The sub-sampling rate was set to 0.5. The network was trained for 50 epochs. Architecture variables are trained using Adam. The learning rate is fixed to $6e-3$. The weight decay is set to $1e-3$. The momentum is set to $(0.5, 0.999)$. In the first 35 epochs, architecture variables are frozen. Network weight parameters are trained using SGD. The initial learning rate is set to 0.5. It is decayed to 0 using cosine scheduling without restart. Momentum is set to 0.9. Weight decay is set to $3e-5$. The batch-size is set to 1024. Epoch number is set to 250. Eight Tesla V100 GPUs were used.

For architecture evaluation on ImageNet, the stacked network starts with three convolution layers which reduce the input image resolution from $224\times224$ to $28\times28$, using stride 2. After the three convolution layers, 12 normal cells and 2 reduction cells are stacked. Initial channel number is set to 48. The network is trained from scratch using SGD for 250 epochs, with batch size 1024, initial learning rate 0.5, weight decay $3e-5$, and momentum 0.9. For the first 5 epochs, learning rate warm-up is used. The learning rate is linearly decayed to 0. Label smoothing and auxiliary loss tower is used.

### M.4 P-DARTS BASED EXPERIMENTS

The search process has three stages. At the first stage, the search space and stacked network in P-DARTS are mostly the same as DARTS. The only difference is the number of cells in the stacked network in P-DARTS is set to 5. At the second stage, the number of cells in the stacked network is 11. At the third stage, the cell number is 17. At stage 1, 2, 3, the initial Dropout probability on skip-connect is 0, 0.4, and 0.7 for CIFAR-10, is 0.1, 0.2, and 0.3 for CIFAR-100; the size of operation space is 8, 5, 3, respectively. The final searched cell is limited to have 2 skip-connect operations at maximum. At each stage, the network is trained using the Adam optimizer for 25 epochs. The batch size is set to 96. The learning rate is set to 6e-4. Weight decay is set to 1e-3. Momentum is set to $(0.5, 0.999)$. In the first 10 epochs, architecture variables are frozen and only network weights are optimized.

For architecture evaluation on CIFAR-100 and CIFAR-10, the stacked network consists of 20 cells. The initial channel number is set to 36. The network is trained from scratch using SGD. The epoch number is set to 600. The batch size is set to 128. The initial learning rate is set to 0.025. The learning rate is decayed to 0 using cosine scheduling. Weight decay is set to 3e-4 for CIFAR-10

and 5e-4 for CIFAR-100. Momentum is set to 0.9. Drop-path probability is set to 0.3. Cutout regularization length is set to 16. Auxiliary towers of weight 0.4 are used.

For architecture evaluation on ImageNet, the settings are similar to those of DARTS. The network consists of 14 cells. The initial channel number is set to 48. The network is trained from scratch using SGD for 250 epochs. Batch size is set to 1024. Initial learning rate is set to 0.5. The learning rate is linearly decayed after each epoch. In the first 5 epochs, learning rate warmup is used. The momentum is set to 0.9. The weight decay is set to $3e-5$. Label smoothing and auxiliary loss tower are used during training. The network was trained on 8 Nvidia Tesla V100 GPUs.

## M.5    PR-DARTS BASED EXPERIMENTS

The operations include: 3×3 and 5×5 separable convolutions, 3×3 and 5×5 dilated separable convolutions, 3×3 average pooling and 3×3 max pooling, zero, and skip connection. The stacked network consists of $k$ cells. The $k/3$- and $2k/3$-th cells are reduction cells. In reduction cells, all operations have a stride of two. The rest cells are normal cells. Operations in normal cells have a stride of one. Cells of the same type (either reduction or normal) have the same architecture. The inputs of each cell are the outputs of two previous cells. Each cell contains four intermediate nodes and one output node. The output node is a concatenation of all intermediate nodes.

For architecture search on CIFAR-100 and CIFAR-10, the stacked network consists of 8 cells. The initial channel number is set to 16. In PR-DARTS, $\lambda_1$, $\lambda_2$, and $\lambda_3$ are set to 0.01, 0.005, and 0.005 respectively. The network was trained for 200 epochs. The mini-batch size is set to 128. Architecture variables are trained using Adam. The learning rate is set to $3e-4$. The weight decay is set to $1e-3$. Network weights are trained using SGD. The initial learning rate is set to 0.025. The momentum is set to 0.9. The weight decay is set to $3e-4$. The learning rate is decayed to 0 using cosine scheduling. For acceleration, per iteration, only two operations on each edge are randomly selected to update. The temperature $\tau$ is set to 10 and is linearly reduced to 0.1; $a = -0.1$ and $b = 1.1$. Pruning on each node is conducted by comparing the gate activation probabilities of all non-zero operations collected from all previous nodes and retaining top two operations.

For architecture evaluation on CIFAR10 and CIFAR100, the stacked network consists of 18 normal cells and 2 reduction cells. The initial channel number is set to 36. The network is trained from scratch using SGD. The mini-batch size is set to 128. The epoch number is set to 600. The initial learning rate is set to 0.025. The momentum is set to 0.9. The weight decay is set to $3e-4$. The gradient norm clipping is set to 5. The drop-path probability is set to 0.2. The cutout length is set to 16. The learning rate is decayed to 0 using cosine scheduling.

For architecture evaluation on ImageNet, the input images are resized to 224×224. The stacked network consists of 3 convolutional layers, 12 normal cells, and 2 reduction cells. The channel number is set to 48. The network is trained using SGD for 250 epochs. The batch size is set to 128. The learning rate is set to 0.025. The momentum is set to 0.9. The weight decay is set to $3e-4$. The gradient norm clipping is set to 5. The learning rate is decayed to 0 via cosine scheduling.

## M.6    HYPERPARAMETER TUNING STRATEGY

To tune the hyperparameters $\lambda$ and $\gamma$, we randomly sample 2.5K data from the 25K training set and sample 2.5K data from the 25K validation set. Then we use the 5K sampled data as a hyperparameter tuning set. $\lambda$ and $\gamma$ are tuned in $\{0.01, 0.1, 0.2, 0.5, 1, 2\}$. For each configuration of $\lambda$ and $\gamma$, we use the remaining 22.5K training data and 22.5K validation data to perform architecture search and use their combination to perform architecture evaluation (retraining a larger stacked network from scratch). Then we measure the performance of the stacked network on the 5K sampled data. $\lambda$ and $\gamma$ yielding the best performance on the 5K sampled data are selected. For other hyperparameters, they mostly follow those in DARTS (Liu et al., 2019a), P-DARTS (Chen et al., 2019), PC-DARTS (Xu et al., 2020), and PR-DARTS (Zhou et al., 2020b).

## M.7 Implementation details

We use PyTorch to implement all models. The version of Torch is 1.4.0 (or above). We build our LPPDT upon official python packages for different differentiable search approaches, such as "DARTS[1]", "P-DARTS[2]" and "PC-DARTS[3]".

## N Additional ablation studies

We perform the following additional ablation study (denoted by NoAux): ignore the auxiliary task entirely, define a quadruple relative similarity (QRS) loss on the validation data, add this loss to the validation loss of the main model, and use the added loss to optimize the architecture.

The results are in Table 22. Our method outperforms NoAux. This shows that the auxiliary task is an important contributor to the effectiveness of our method.

| Method | Test error on CIFAR100 | Test error on CIFAR10 |
|---|---|---|
| NoAux+Darts2nd | 19.96±0.15 | 2.88±0.08 |
| Ours+Darts2nd | **17.21**±0.10 | **2.62**±0.06 |
| NoAux+Pdarts | 19.62±0.06 | 2.60±0.04 |
| Ours+Pdarts | **16.05**±0.08 | **2.47**±0.03 |

Table 22: Results of the NoAux ablation study.

## O Experimental details of results in Figure 2 in the main paper

First, we create a collection $C$ of quadruples which have correct relative similarity relationships. We randomly sample two data instances $x$ and $y$ from the same class and randomly sample two data instances $z$ and $w$ from different classes. Since $x$ and $y$ are from the same class, their similarity is considered as being larger than that between $z$ and $w$ which are from different classes. Then we put $(x,y) \succ (z,w)$ into $C$. This procedure repeats 10K times. In the end, $C$ contains 10K quadruples. Then from $C$, we create another three collections of quadruples: $C_{25}$, $C_{50}$, and $C_{75}$, with increasing levels of incorrect quadruple relative similarity relationships. To get $C_{25}$, we randomly sample 25% quadruples from $C$; then for each sampled quadruple $(x,y) \succ (z,w)$, we switch the order of these two pairs and get $(z,w) \succ (x,y)$, which is an incorrect quadruple. In $C_{25}$, 25% quadruples have incorrect relative similarities. $C_{50}$ and $C_{75}$ are obtained in similar ways.

Given $C_{25}$, we use it to train an auxiliary model, by solving the following problem:

$$\min_A L_a(A, E_a^*(A), H_a^*(A), D_a^{(val)})$$
$$s.t. \ E_a^*(A), H_a^*(A) = \operatorname{argmin}_{E_a, H_a} L_a(A, E_a, H_a, D_a^{(tr)}) + \tag{22}$$
$$\lambda \sum_{(x,y) \succ (z,w) \in C_{25}} \max(0, -(c(x,y;A,E_a) - c(z,w;A,E_a)))$$

Then we measure the classification error of this trained auxiliary model on the validation set $D_a^{(val)}$ of the auxiliary. Similarly, we do this for $C_{50}$ and $C_{75}$. The classification errors are shown as the blue curve in Figure 2 in the main paper.

Next, we discuss how to obtain the orange curve. Given $C_{25}$, we use it to train a main model, by solving the following problem:

$$\min_M \sum_{(x,y) \succ (z,w) \in C_{25}} \max(0, -(c(x,y;M,E_m^*(M)) - c(z,w;M,E_m^*(M))))$$
$$s.t. \ E_m^*(M), H_m^*(M) = \operatorname{argmin}_{E_m, H_m} L_m(M, E_m, H_m, D_m^{(tr)}) \tag{23}$$

Then we measure the classification error of this trained main model on the validation set $D_m^{(val)}$ of the main model. Similarly, we do this for $C_{50}$ and $C_{75}$. The classification errors are shown as the orange curve in Figure 2 in the main paper.

---

[1] https://github.com/quark0/darts

[2] https://github.com/chenxin061/pdarts

[3] https://github.com/yuhuixu1993/PC-DARTS/

## P    EXPERIMENTAL DETAILS OF RESULTS IN FIGURE 2 IN THE MAIN PAPER

In the pseudo-labeling (PS) baseline, the main model and auxiliary model predict class probabilities (pseudo labels) on the same images, then a KL divergence is defined on these probabilities and the auxiliary model is trained by minimizing the KL divergence. The setup follows equation 3 in (Zhang et al., 2018).

For the pairwise similarity (PS) baseline, the formalism is: given two images, the main model and auxiliary model predict the probabilities regarding whether these two images are similar, then a cross-entropy loss is defined on these two probabilities; the auxiliary model is trained by minimizing this cross-entropy loss.

For the triple-wise similarity (TS) baseline, given three data examples $x$, $y$, $z$, the main model predicts whether $x$ is more close to $y$ or to $z$; the auxiliary model is trained by minimizing a hinge loss that is consistent with the prediction of the main model on this triple. The formalism follows equation 6 in (You et al., 2017).

## Q    EXPERIMENTAL DETAILS OF EVALUATING ROBUSTNESS AGAINST OVERFITTING

The four search spaces $S1 - S4$ are designed by (Zela et al., 2020).

- **S1**: In this search space, each edge has only two candidate operations. To identify these operations, operations that have the least importance in the original search space of DARTS are iteratively removed.
- **S2**: For each edge, the candidate operations are 3×3 SepConv and SkipConnect.
- **S3**: For each edge, the candidate operations are: 3×3 SepConv, SkipConnect, and Zero.
- **S4**: For each edge, the candidate operations are: 3×3 SepConv and Noise. In the Noise operation, every value from the input feature map is replaced with random variables sampled from univariate Gaussian distribution.

## R    SIGNIFICANCE TEST RESULTS

To check whether the performance of our proposed methods are significantly better than baselines, we perform a statistical significance test using a double-sided T-test. We use the function in the python package "scipy.stats.ttest_1samp" and report the average results over 10 different runs. Table 23 and 24 show the results.

From these two tables, we can see that the p-values are small between baselines methods and our methods, which demonstrate that the errors of our methods are significantly lower than those of baselines.

## S    MODEL PARAMETERS, SEARCH COSTS, AND FLOPS ON IMAGENET

Table 25 shows the number of model parameters, search costs, and FLOPs on ImageNet. The parameter numbers, search costs, and FLOPs of our methods are close to those in differentiable baselines.

## T    FULL LISTS OF HYPERPARAMETER SETTINGS IN NAS EXPERIMENTS

The hyperparameter settings for the auxiliary model such as the optimizer, momentum, weight decay, learning rate, number of layers were the same as those of the original DARTS and PC-DARTS implementations. The hyperparameter settings for the main model were also set to the same values as those of the auxiliary model. Tables 26-29 show the hyperparameter settings used in architecture search experiments. Tables 30-33 show the hyperparameter settings used in architecture evaluation experiments.

| Our method | Baseline | p-value |
|---|---|---|
| TETLO-darts2nd | MTL-darts2nd | 9.31e-5 |
| TETLO-darts2nd | ES-darts2nd | 2.66e-8 |
| TETLO-darts2nd | ATT-darts2nd | 7.05e-9 |
| TETLO-darts2nd | Darts2nd | 4.49e-11 |
| TETLO-pdarts | MTL-pdarts | 3.92e-5 |
| TETLO-pdarts | ES-pdarts | 6.30e-6 |
| TETLO-pdarts | ATT-pdarts | 7.19e-6 |
| TETLO-pdarts | Pdarts | 4.82e-7 |
| TETLO-pcdarts | MTL-pcdarts | 5.38e-6 |
| TETLO-pcdarts | ES-pcdarts | 7.21e-6 |
| TETLO-pcdarts | ATT-pcdarts | 4.63e-6 |
| TETLO-pcdarts | Pcdarts | 8.31e-7 |
| TETLO-prdarts | MTL-prdarts | 1.65e-7 |
| TETLO-prdarts | ES-prdarts | 3.07e-6 |
| TETLO-prdarts | ATT-prdarts | 4.77e-7 |
| TETLO-prdarts | Prdarts | 5.28e-6 |

Table 23: Significance test results on CIFAR-100

| Our method | Baseline | p-value |
|---|---|---|
| TETLO-darts2nd | MTL-darts2nd | 5.02e-9 |
| TETLO-darts2nd | ES-darts2nd | 7.39e-10 |
| TETLO-darts2nd | ATT-darts2nd | 6.20e-10 |
| TETLO-darts2nd | Darts2nd | 3.51e-10 |
| TETLO-pdarts | MTL-pdarts | 4.09e-7 |
| TETLO-pdarts | ES-pdarts | 6.83e-8 |
| TETLO-pdarts | ATT-pdarts | 8.04e-8 |
| TETLO-pdarts | Pdarts | 6.37e-7 |
| TETLO-pcdarts | MTL-pcdarts | 7.26e-10 |
| TETLO-pcdarts | ES-pcdarts | 3.51e-10 |
| TETLO-pcdarts | ATT-pcdarts | 5.05e-10 |
| TETLO-pcdarts | Pcdarts | 6.82e-10 |
| TETLO-prdarts | MTL-prdarts | 6.93e-11 |
| TETLO-prdarts | ES-prdarts | 3.85e-12 |
| TETLO-prdarts | ATT-prdarts | 7.22e-7 |
| TETLO-prdarts | Prdarts | 3.59e-5 |

Table 24: Significance test results on CIFAR-10

Table 25: Top-1 and top-5 classification errors on ImageNet test set, number of model weights (millions), and search cost (GPU days), and FLOPs (M). TETLO-darts2nd-rn50-cifar10 denotes an architecture searched by applying TETLO (with ResNet-50 as main encoder) to DARTS-2nd on CIFAR-10. Similar meanings hold for other notations having such a format. Results marked with * are taken from DARTS$^-$ (Chu et al., 2020a) and DrNAS (Chen et al., 2020b). The other notations are the same as those in Table 2 in the main paper.

| Method | Top-1 Error (%) | Top-5 Error (%) | Param (M) | Cost (GPU days) | FLOPs (M) |
|---|---|---|---|---|---|
| *Inception-v1 (Szegedy et al., 2015) | 30.2 | 10.1 | 6.6 | - | 1448 |
| *ShuffleNet 2× (v2) (Ma et al., 2018) | 25.1 | 7.6 | 7.4 | - | 299 |
| *NASNet-A (Zoph et al., 2018) | 26.0 | 8.4 | 5.3 | 1800 | 564 |
| *AmoebaNet-C (Real et al., 2019) | 24.3 | 7.6 | 6.4 | 3150 | 570 |
| *SDARTS-ADV-CIFAR10 (Chen & Hsieh, 2020) | 25.2 | 7.8 | 5.4 | 1.3 | - |
| *PC-DARTS-CIFAR10 (Xu et al., 2020) | 25.1 | 7.8 | 5.3 | 0.1 | 586 |
| *ProxylessNAS-ImageNet (Cai et al., 2019) | 24.9 | 7.5 | 7.1 | 8.3 | 465 |
| *FairDARTS-ImageNet (Chu et al., 2019) | 24.4 | 7.4 | 4.3 | 3.0 | 440 |
| *DARTS$^+$-CIFAR100 (Liang et al., 2019) | 23.7 | 7.2 | 5.1 | 0.2 | 591 |
| *Darts2nd-cifar10 (Liu et al., 2019a) | 26.7 | 8.7 | 4.7 | 4.0 | 574 |
| ATT-darts2nd-cifar10 | 26.3 | 8.2 | 4.8 | 4.4 | 539 |
| ES-darts2nd-cifar10 | 26.4 | 8.5 | 4.8 | 4.0 | 582 |
| MTL-darts2nd-cifar10 | 26.4 | 8.3 | 4.7 | 4.2 | 546 |
| TETLO-darts2nd-cifar10 (ours) | **25.4** | **7.8** | 4.6 | 4.0 | 551 |
| *Pdarts-cifar10 (Chen et al., 2019) | 24.4 | 7.4 | 4.9 | 0.3 | 557 |
| ATT-pdarts-cifar10 | 24.4 | 7.2 | 4.9 | 0.5 | 583 |
| ES-pdarts-cifar10 | 24.4 | 7.3 | 5.0 | 0.3 | 526 |
| MTL-pdarts-cifar10 | 24.3 | 7.2 | 4.8 | 0.4 | 559 |
| TETLO-pdarts-cifar10 (ours) | **23.6** | **7.0** | 4.8 | 0.3 | 538 |
| *Pdarts-cifar100 (Chen et al., 2019) | 24.7 | 7.5 | 5.1 | 0.3 | 577 |
| ATT-pdarts-cifar100 | 24.6 | 7.5 | 5.1 | 0.5 | 549 |
| ES-pdarts-cifar100 | 24.6 | 7.5 | 5.3 | 0.3 | 552 |
| MTL-pdarts-cifar100 | 24.7 | 7.7 | 5.1 | 0.4 | 561 |
| TETLO-pdarts-cifar100 (ours) | **24.1** | **7.3** | 5.1 | 0.3 | 531 |
| *Pcdarts-imagenet (Xu et al., 2020) | 24.2 | 7.3 | 5.3 | 3.8 | 597 |
| ATT-pcdarts-imagenet | 24.0 | 7.1 | 5.5 | 4.3 | 583 |
| ES-pcdarts-imagenet | 24.1 | 7.2 | 5.2 | 3.8 | 542 |
| MTL-pcdarts-imagenet | 23.9 | 7.3 | 5.5 | 3.9 | 566 |
| TETLO-pcdarts-imagenet (ours) | **23.3** | **6.2** | 5.3 | 4.0 | 574 |

| Name | Value |
|---|---|
| Optimizer | SGD |
| Learning rate of architecture variables | 3e-4 |
| Epochs | 30 |
| Weight decay of architecture variables | 1e-3 |
| Batch size | 32 |
| Drop path probability | 3e-1 |
| Initial channels | 16 |
| Learning rate of network weights | 2.5e-2 |
| Number of layers | 8 |
| Momentum | 9e-1 |
| Weight decay for network weights | 3e-4 |
| Gradient clip | 5 |
| Lambda | 0.1 |
| Train portion | 0.5 |
| Unrolled | True |
| Cutout | False |

Table 26: Hyperparameter settings of our TETLO method when applied to DARTS, during architecture search on CIFAR-10 and CIFAR-100.

| Name | Value |
|------|-------|
| Optimizer | SGD |
| Learning rate of architecture variables | 6e-4 |
| Epochs | 50 |
| Weight decay of architecture variables | 1e-3 |
| Batch size | 96 |
| Drop path probability | 3e-1 |
| Initial channels | 16 |
| Learning rate of network weights | 1e-1 |
| Number of layers | 8 |
| Momentum | 9e-1 |
| Weight decay for network weights | 3e-4 |
| Grad clip | 5 |
| Lambda | 0.5 |
| Train portion | 0.5 |
| Unrolled | True |
| Cutout | False |

Table 27: Hyperparameter settings of our TETLO method when applied to PC-DARTS, during architecture search on CIFAR-10 and CIFAR-100.

| Name | Value |
|------|-------|
| Optimizer | SGD |
| Learning rate of architecture variables | 3e-4 |
| Epochs | 30 |
| Weight decay of architecture variables | 1e-3 |
| Batch size | 32 |
| Drop path probability | 3e-1 |
| Initial channels | 16 |
| Learning rate of network weights | 2.5e-2 |
| Number of layers | 8 |
| Momentum | 9e-1 |
| Weight decay for network weights | 3e-4 |
| Gradient clip | 5 |
| Lambda | 0.1 |
| Train portion | 0.5 |
| Unrolled | True |
| Cutout | False |

Table 28: Hyperparameter settings of our TETLO method when applied to PDARTS, during architecture search on CIFAR-10 and CIFAR-100.

| Name | Value |
|---|---|
| Optimizer | SGD |
| Learning rate of architecture variables | 6e-4 |
| Epochs | 50 |
| Weight decay of architecture variables | 1e-3 |
| Batch size | 96 |
| Drop path probability | 3e-1 |
| Initial channels | 16 |
| Learning rate of network weights | 1e-1 |
| Number of layers | 8 |
| Momentum | 9e-1 |
| Weight decay for network weights | 3e-4 |
| Grad clip | 5 |
| Lambda | 0.5 |
| Train portion | 0.5 |
| Unrolled | True |
| Cutout | False |

Table 29: Hyperparameter settings of our TETLO method when applied to PR-DARTS, during architecture search on CIFAR-10 and CIFAR-100.

| Name | Value |
|---|---|
| Optimizer | SGD |
| Learning rate | 2.5e-2 |
| Epochs | 600 |
| Weight decay | 3e-4 |
| Batch size | 96 |
| Momentum | 9e-1 |
| Initial channels | 36 |
| Number of layers | 20 |
| Auxiliary | True |
| Auxiliary weight | 0.4 |
| Cutout | True |
| Cutout length | 16 |
| Drop path probability | 0.2 |
| Gradient clip | 5 |

Table 30: Hyperparameter settings of our TETLO method when applied to DARTS, during architecture evaluation on CIFAR-10 and CIFAR-100.

| Name | Value |
|---|---|
| Optimizer | SGD |
| Learning rate | 2.5e-2 |
| Epochs | 600 |
| Weight decay | 3e-4 |
| Batch size | 96 |
| Momentum | 9e-1 |
| Initial channels | 36 |
| Number of layers | 20 |
| Auxiliary | True |
| Auxiliary weight | 0.4 |
| Cutout | True |
| Cutout length | 16 |
| Drop path probability | 0.3 |
| Gradient clip | 5 |

Table 31: Hyperparameter settings of our TETLO method when applied to PC-DARTS, during architecture evaluation on CIFAR-10 and CIFAR-100.

| Name | Value |
|---|---|
| Optimizer | SGD |
| Learning rate | 2.5e-2 |
| Epochs | 600 |
| Weight decay | 3e-4 |
| Batch size | 96 |
| Momentum | 9e-1 |
| Initial channels | 36 |
| Number of layers | 20 |
| Auxiliary | True |
| Auxiliary weight | 0.4 |
| Cutout | True |
| Cutout length | 16 |
| Drop path probability | 0.2 |
| Gradient clip | 5 |

Table 32: Hyperparameter settings of our TETLO method when applied to PDARTS, during architecture evaluation on CIFAR-10 and CIFAR-100.

| Name | Value |
|---|---|
| Optimizer | SGD |
| Learning rate | 2.5e-2 |
| Epochs | 600 |
| Weight decay | 3e-4 |
| Batch size | 96 |
| Momentum | 9e-1 |
| Initial channels | 36 |
| Number of layers | 20 |
| Auxiliary | True |
| Auxiliary weight | 0.4 |
| Cutout | True |
| Cutout length | 16 |
| Drop path probability | 0.3 |
| Gradient clip | 5 |

Table 33: Hyperparameter settings of our TETLO method when applied to PR-DARTS, during architecture evaluation on CIFAR-10 and CIFAR-100.

## U    VISUALIZATION OF SEARCHED ARCHITECTURES

Using the graphviz package, we plot the searched architecture cells in all the experiments. Each architecture consists of a normal cell and a reduction cell. Figure 7-14 show the cells searched on CIFAR-10 and CIFAR-100 by our TETLO methods.

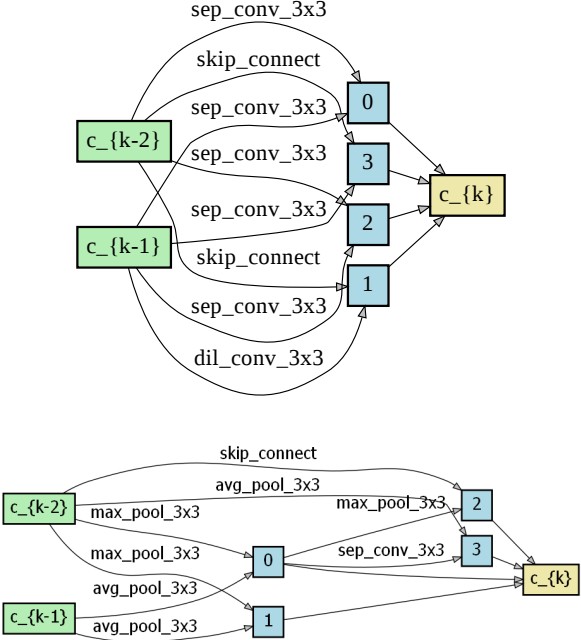

Figure 7: Normal cell (top) and reduction cell (bottom) searched by TETLO-DARTS on CIFAR-100.

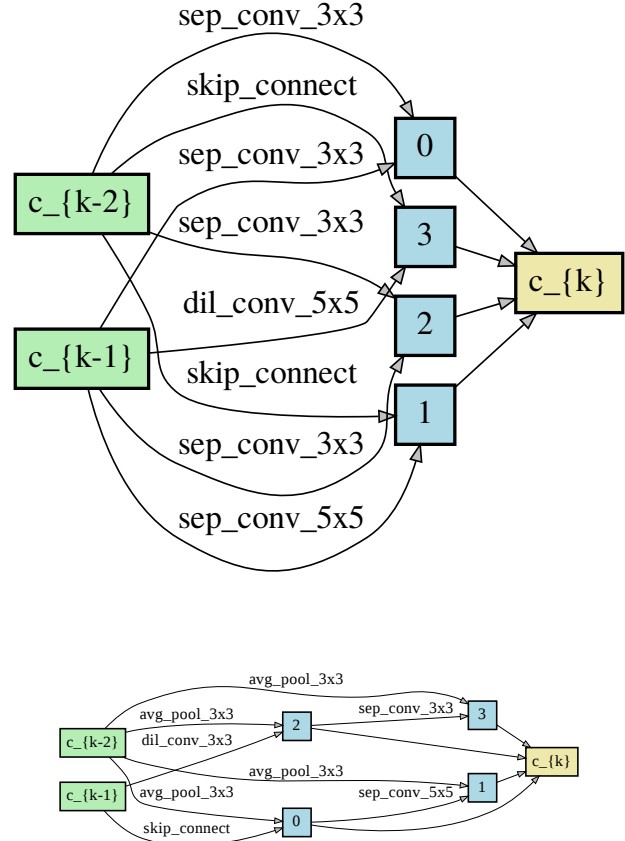

Figure 8: Normal cell (top) and reduction cell (down) searched by TETLO-DARTS on CIFAR-10.

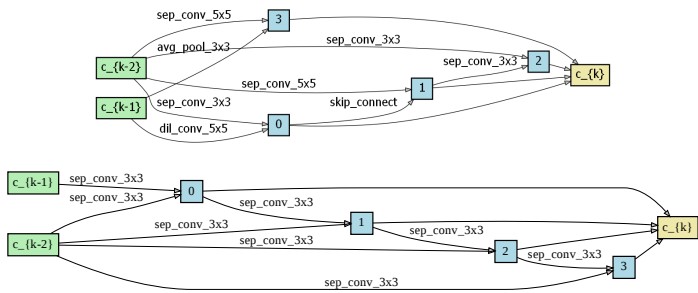

Figure 9: Normal cell (top) and reduction cell (bottom) searched by TETLO-PDARTS on CIFAR-100.

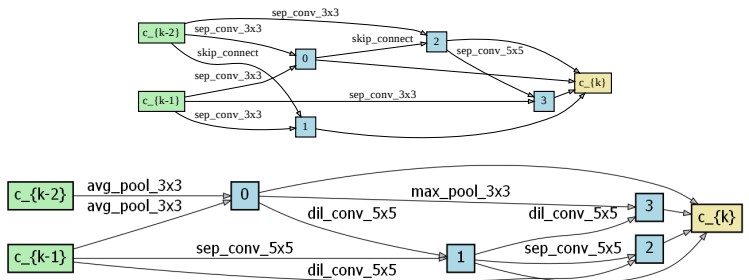

Figure 10: Normal cell (top) and reduction cell (bottom) searched by TETLO-PDARTS on CIFAR-10.

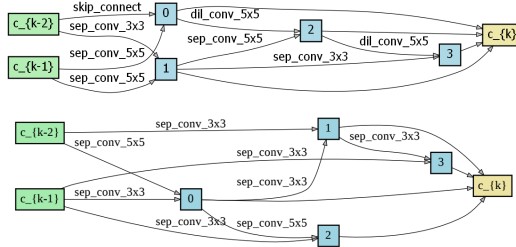

Figure 11: Normal cell (top) and reduction cell (bottom) searched by TETLO-PCDARTS on CIFAR-100.

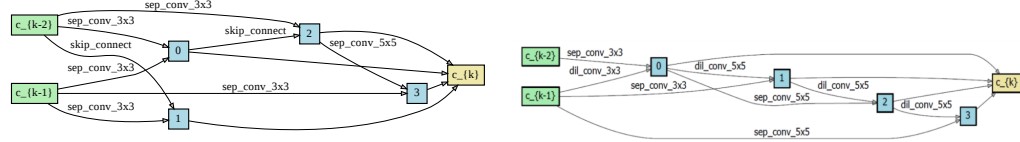

Figure 12: Normal cell (left) and reduction cell (right) searched by TETLO-PCDARTS on CIFAR-10.

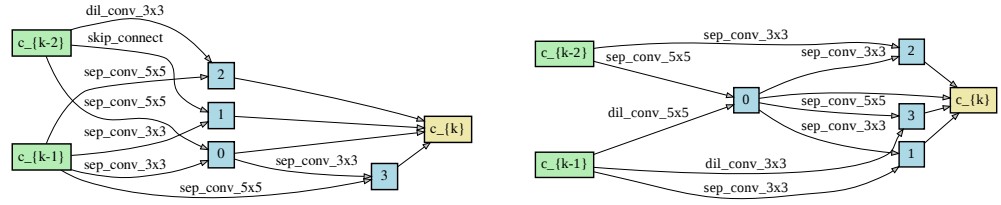

Figure 13: Normal cell (left) and reduction cell (right) searched by TETLO-PRDARTS on CIFAR-100.

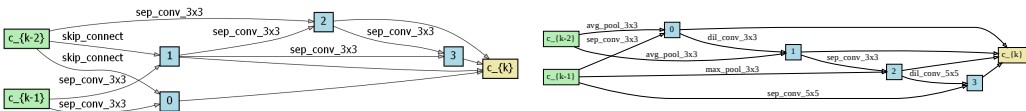

Figure 14: Normal cell (left) and reduction cell (right) searched by TETLO-PRDARTS on CIFAR-10.

## V FORMULATIONS OF BASELINES

For multi-task learning (MTL), the formulation is:

$$
\min_{M,A} L_m(M, E_m^*(M,A), H_m^*(M), D_m^{(val)}) + \gamma L_a(A, E_a^*(M,A), H_a^*(A), D_a^{(val)})
$$
$$
s.t.
$$
$$
E_m^*(M,A), H_m^*(M), E_a^*(M,A), H_a^*(A) = \mathrm{argmin}_{E_m, H_m, E_a, H_a} L_a(A, E_a, H_a, D_a^{(tr)}) +
$$
$$
L_m(M, E_m, H_m, D_m^{(tr)}) + \lambda \sum_{x,y \in D_a^{(tr)}} \max(0, -(c(x,y;E_m,M) - c(z,w;E_m,M))(c(x,y;E_a,A) - c(z,w;E_a,A)))
$$
$$
\tag{24}
$$

For encoder sharing (ES), the formulation is:

$$
\min_A L_m(M, E^*(M), H_m^*(M), D_m^{(val)}) + \gamma L_a(M, E^*(M), H_a^*(M), D_a^{(val)})
$$
$$
s.t. \quad E^*(M), H_m^*(M), H_a^*(M) = \mathrm{argmin}_{E, H_a, H_m} L_a(M, E, H_a, D_a^{(tr)}) + L_m(M, E, H_m, D_m^{(tr)})
$$
$$
\tag{25}
$$

where $E$ is the shared encoder.

## W ASSUMPTION ON TRANSFER LEARNING

Since our proposed transfer learning method is based on predicting quadruple relative similarities, the assumption is that the relative similarities between data examples are unchanged between domains. In addition, we assume the input data modality and the output label space are uncharged between domains.

We expect the main and auxiliary tasks to be similar, to ensure the knowledge can be successfully transferred from the main model to the auxiliary model. If the two tasks are largely dissimilar, the auxiliary may not be able to help the main model to improve.

## X ADDITIONAL RELATED WORKS

**Bi-level optimization.** Our work is based on tri-level optimization, which is related to bi-level optimization (BLO). BLO (Dempe, 2002) has been broadly applied for hyperparameter tuning (Feurer et al., 2015), neural architecture search (Liu et al., 2019a), meta learning (Finn et al., 2017), data reweighting (Shu et al., 2019; Ren et al., 2020; Wang et al., 2020), learning rate adjustment (Baydin et al., 2017), label denoising (Zheng et al., 2019), data generation (Such et al., 2019). In these methods, meta parameters (e.g, hyperparameters, neural architectures, data weights, etc.) are optimized by minimining validation losses and model weights are learned by minimizing training losses.

**Multi-task learning (MTL).** MTL (Ruder, 2017; Zhang & Yang, 2021) aims to improve multiple tasks simultaneously by learning them jointly and transferring knowledge across tasks. Various MTL approaches have been proposed, based on 1) hard parameter sharing (Caruana, 1998; Long et al., 2015; Doersch & Zisserman, 2017; Kokkinos, 2017; Sener & Koltun, 2018; Leang et al., 2020), where multiple tasks share the same weight parameters, such as encoder weights; 2) soft parameter sharing (Duong et al., 2015; Dai et al., 2016; Misra et al., 2016; Yang & Hospedales, 2016; Lu et al., 2017; Liu et al., 2019c; Maninis et al., 2019), where parameters of different tasks are constrained to be similar; 3) task similarity learning (Williams et al., 2007; Zhang & Yeung, 2014; Bingel & Søgaard, 2017; Standley et al., 2020; Zamir et al., 2020), which identifies similarity between tasks and encourages similar tasks to share more commonalities; 4) loss weighting (Chen et al., 2018; Kendall et al., 2018; Sener & Koltun, 2018; Gong et al., 2019; Leang et al., 2020), which weighs each task's loss. Our work is also related to meta learning, which learns to transfer meta knowledge to end tasks (Lee & Choi, 2018; Yao et al., 2020; Hospedales et al., 2020; Andrychowicz et al., 2016; Li & Malik, 2016; Ravi & Larochelle, 2016; Finn et al., 2017; Li et al., 2017; Antoniou et al., 2018; Mishra et al., 2017; Qiao et al., 2018; Gidaris & Komodakis, 2018; Lekkala & Itti, 2020; Sun et al., 2020).

**Neural architecture search (NAS).** NAS aims to automatically identify highly-performing architectures of deep neural networks instead of manually designing them by humans. Various approaches have been proposed for NAS, including differentiable search methods (Cai et al., 2019;

Liu et al., 2019a; Xie et al., 2019) and those based on reinforcement learning (Zoph & Le, 2017; Pham et al., 2018; Zoph et al., 2018) and evolutionary algorithms (Liu et al., 2018b; Real et al., 2019). In RL-based approaches, a policy is learned to iteratively generate new architectures by maximizing a reward which is the accuracy on the validation set. Evolutionary algorithm based approaches represent architectures as individuals in a population. Individuals with high fitness scores (validation accuracy) have the privilege to generate offspring, which replaces individuals with low fitness scores. Differentiable approaches adopt a network pruning strategy. On top of an over-parameterized network, importance weights of building blocks are learned using gradient descent. After learning, blocks whose weights are close to zero are pruned. There have been many efforts devoted to improving differentiable NAS methods. In P-DARTS (Chen et al., 2019), the depth of searched architectures is allowed to grow progressively during the training process. Search space approximation and regularization approaches are developed to reduce computational overheads and improve search stability. PC-DARTS (Xu et al., 2020) reduces the redundancy in exploring the search space by sampling a small portion of a super network. Operation search is performed in a subset of channels with the held-out part bypassed in a shortcut.

