# OpenReview forum: "Improving Differentiable Neural Architecture Search by Encouraging Transferability"
_ICLR.cc/2023/Conference — ICLR 2023 poster_

### Official Review · Reviewer_3zEW · 2022-10-25

**Confidence:** 2
**Clarity, Quality, Novelty And Reproducibility:** 1. **Clarity**
**Correctness:** 3
**Technical Novelty And Significance:** 3
**Empirical Novelty And Significance:** 3
**Recommendation:** 8

**Strength And Weaknesses:**

## Strengths
1. Although the content of the article is a bit large, it is generally organized and the article is easy to understand. The formulas are also deduced in detail.
2. The experimental part is solid and has achieved good performance.
3. The way to solve differentiable NAS by introducing an auxiliary task is innovative.

## Weaknesses
There are two points I don't quite understand:
1. Why use a method based on matching quadruple relative similarities for knowledge transfer? What other options are there to do this, and why is this way better than others? I don't think the insight behind this is sufficient.
2. In this paper, the stability of the method is proved by the experiments in Table 8. The stability I understand should be the frequency of network structure degradation in a large number of experiments, not the number of skip-connections in the network. I don't know if there is a more reasonable way to prove this.

**Summary Of The Paper:**

This paper introduces a method to improve the stability of differentiable NAS. It can improve stability and performance without introducing too much cost.

The major contributions of this work include:
+ We propose a transferability-encouraging tri-level optimization framework to improve
the generalizability and stability of differentiable NAS methods.
+ We propose a new knowledge transfer approach based on matching quadruple relative similarities.
+ We perform various experiments which demonstrate the effectiveness of our method.


**Summary Of The Review:**

I think this is an informative article. The solution is more reasonable, and a large number of experiments are used to support the conclusions in the text.

---

### Official Review · Reviewer_8tqH · 2022-10-26

**Confidence:** 3
**Correctness:** 3
**Technical Novelty And Significance:** 3
**Empirical Novelty And Significance:** 3
**Recommendation:** 5

**Clarity, Quality, Novelty And Reproducibility:**

The paper has a good clarity. But some important details need more description, as listed in the weaknesses section.

**Strength And Weaknesses:**

Strengths:

This paper develops a transferability-encouraging tri-level optimization framework in differentiable NAS and demonstrates that they can achieve higher performance on Imagenet and CIFAR compared to similar methods.

The authors introduce a new knowledge transfer approach based on matching quadruple relative similarities (QRS) to improve the auxiliary model and demonstrate its effects with ablations.

The paper is well written and easy to understand.

Weaknesses:

1. The method consists of training the main model, training the auxiliary model with the help of the main model (to determine similar data pair), and optimization. The main contribution seems to be the auxiliary model using the data pairs to learn the weights. In other words, the fixed main model provides the supervisory signal for the auxiliary model. The main methodology does not seem to be novel.

2. Most of the baseline methods in Table 3&4 are at least 2 years ago. The work lacks comparison with more latest methods in experiments.

3. Are the two data pairs (x,y) and (z,w) randomly sampled?

4. In the case of cosine similarity c(x,y) > c(z,w), it appears that x,y,z and w examples are 4 class examples. In this case, how to optimize them?

5. What is the difference between the data D_a and D_m?

**Summary Of The Paper:**

This paper proposes to improve the generalizability and stability of differentiable NAS methods. Specifically, the authors first train a main model and then transfer knowledge from the main model to an auxiliary model. Finally, the authors optimize the architecture of the main model by maximizing its transferability to the auxiliary model. Experiments on several datasets show the effectiveness of the proposed model.

**Summary Of The Review:**

Although the proposed method is simple, the solution is somewhat trivial (e.g., quadruple relative similarity relation using cosine similarity strategy). The experiments need comparison with more latest methods.

---

> ### Comment · Reviewer_8tqH · 2022-12-07
> **thanks for the response**
>
> Thanks for the detailed response and the revision of the paper. It solves most of my concerns and questions.
>
> But for the novelty, I have not been fully convinced. I think the authors should highlight why generalizability is a significant problem for NAS. I think the solved limitations are not common drawbacks in existing NAS methods. For example, not all methods suffer from degenerate architecture and cannot control the number of skip connections. Why is it necessary for the learned representation to be transferable to other models? What is its rigorous connection with the number of skip connections? So, I do not think the solved limitations are significant for NAS. Besides, the necessity of the introduced tri-level optimization for NAS is also confusing to me. In DARTS, the second-order optimization induces much computation cost, so they also suggest the one-order optimization. There is also a study (arXiv:2110.04743) that proposes the zero-order DARTS. But the authors of this paper propose a higher order, tri-level optimization framwork. It confuses me whether lower order or higher order is better for NAS. Intuitively, higher level objective increases the instability and difficulty for optimization.
>
> So, I tend to keep my original score. But I will not object if this paper is accepted.

---

### Official Review · Reviewer_5nMq · 2022-10-28

**Confidence:** 4
**Correctness:** 4
**Technical Novelty And Significance:** 4
**Empirical Novelty And Significance:** 4
**Recommendation:** 8

**Clarity, Quality, Novelty And Reproducibility:**

This paper is of good writing quality.All the ideas and points are easy to follow.
The idea is somehow novel.
The evaluation is good and enough to prove the effacy of the method.

**Strength And Weaknesses:**

# Strength
- S1.The paper solves the stability issues of DARTS from a  different aspect, transfer learning view, which is somehow novel and interesting. The motivation is reasonable for me.
- S2.The paper is well-written and easy to follow.
- S3.The evaluations are sufficient to prove the validity of the proposed methods.
- S4.The results are promising on C10, C100, and ImageNet.

# Weaknesses
- Q1.What are the computation complexity and extra cost when applying QRS compared with second-            order or third-order? Intuitively, when increasing the order, the training cost may increase either. Ablations about this should be added.
- Q2.In Page 5, the author claims that the auxiliary model ( encoder) is the same as the main model which is a DARTS. It is clear that the main model weights are trained in the first stage and the alpha is trained in the third stage. But it is unclear in the second stage. how to optimize the auxiliary model. Whether to fix the alpha and only train the weights or alternatively train the weights and alpha just like the original DARTS[1], with the head changed to predict the correlations between two pairs of examples. The author should clarify this procedure.
- Q3.As shown in Page 5, the author ties the architectures and weights of main and auxiliary models.  In what stage they are tied? And my concern is, if the main and the auxilary are the same model both in weights and alpha, actually there are no knowledge trainsfer but only model copy. Why a copied model can take effect in the selection of the alpha?
- Q4.Is it necessary that auxiliary model have to be the DARTS? Does the author try any other auxiliary mode, such as ResNet50?
- Q5.Further, maybe the choice of the auxiliary model would be the bottelneck of the method. Maybe more ablations about how the size of the auxilary model affect the main model is needed. Such as, choose ResNet18, ResNet50 and ResNet101 or DARTS with 1,3,5,8 choice blocks as auxiliary model.
- Q6. Just curious, what is the results when applying on DARTS 1st order?
- Q7.The method may be too complex and hand-crafted
- Q8.What is the generalization performance when search a model from C100 and evaluate it on C10 and search from C10 and evaluate on C100.


**Summary Of The Paper:**

The author adopts knowledge transfer to solve the inhabited stability and generalization issue of differential NAS. The method first trains the weights of the main model with alpha fixed, then transfer the knowledge from main to auxiliary by quadruple relative similarities, and finally optimizes the alpha of the main model by optimizing the transferability to the auxiliary.
The intuition behind this is that, once the architecture equips with a good architecture that can extract useful knowledge, the obtained knowledge is accurate and good enough to transfer to the auxiliary one, and the auxiliary reaches high quality.

**Summary Of The Review:**

Overall, this is a good paper. The writing is clear and the evaluation is good enough. The results show its effacy.

# Update

Since my questions are well answered, I have no further comments and raise my score to 8.

---

### Official Review · Reviewer_8AHX · 2022-11-01

**Confidence:** 3
**Correctness:** 2
**Technical Novelty And Significance:** 2
**Empirical Novelty And Significance:** 2
**Recommendation:** 6

**Clarity, Quality, Novelty And Reproducibility:**

There exist few points to be clarified for improving the quality and originality, as mentioned above.

**Strength And Weaknesses:**

**Strength**
- The paper is easy to read.
- They tackle the practical limitations of neural architecture search.
- Experiments are extensive.

**Weaknesses**
- My main concern for this paper is the quality of the validation sets, which play a key role in measuring “transferability” from the main model to the auxiliary models. The poor representativeness of validation sets (class-imbalanced, noise, etc) for the main model and auxiliary model may affect the optimization process (step 3) and this should be clearly analyzed. What if we cannot get the class-balanced and non-noisy validation set in a practical scenario? The proposed method with a poor validation set still outperforms the baseline models without any degeneration? If not, are there any possible methods to handle this issue?
- For one of the baseline models, OFA-large (Table 4) seems to show the best performance, but I was not able to find any explanation for it. As far as I know, OFA can sample subnets from the supernet while keeping (or transferring) the supernet’s pretrained knowledge, which is similar to the proposed method. It would be good to compare the proposed method with OFA more closely.
- (minor) It would be helpful if the overview illustration for the proposed method is provided.


**Summary Of The Paper:**

The authors tackle the current limitation that most existing NAS methods suffer from, which are unsatisfactory generalizability and stability, such as generating a dominant number of skip connections or poor test performance. To address this, the authors propose a transferability-encouraging tri-level optimization framework for improving the main model and auxiliary models. They demonstrate the effectiveness of their methods.

**Summary Of The Review:**

I enjoyed reading the paper, but several improvements seem to be required.

---

### Decision · Program_Chairs · 2023-01-20

**Decision:**

Accept: poster

**Justification For Why Not Higher Score:**

Respecting the reservations from Reviewer 8tqH on the paper

**Justification For Why Not Lower Score:**

Most reviewers tend to agree on acceptance, except reviewer 8tqH who has a weak reject score but the meta-reviewer believes the authors have addressed the last three questions from 8tqH well.

**Metareview: Summary, Strengths And Weaknesses:**

This paper proposes a method to improve the generalizability and stability of differentiable NAS methods. Specifically, the authors propose to encourage the transferability of the learned architecture through a tri-level optimization framework that first trains the main model, transfers knowledge from the main model to an auxiliary model, and then optimizes the architecture of the main model by maximizing its transferability to the auxiliary model. Three of the reviewers appreciated that author response addressed their concerns quite well. There were some follow-up questions from Reviewer 8tqH on the importance of generalizability and transferability in NAS, and on high-level vs low-level optimization in NAS which the meta-reviewer believes the authors have addressed.

**Note From Pc:**

if the above contains the word "oral" or "spotlight" please see: "oral" presentation means -> notable-top-5% and "spotlight" means -> notable-top-25%. As stated in our emails, we are disassociating presentation type from AC recommendations